# Neural representations of honesty predict future trust behavior

Gabriele Bellucci [1,2,3]*, Felix Molter [4] & Soyoung Q. Park[1,3,5,6]

Theoretical accounts propose honesty as a central determinant of trustworthiness impressions and trusting behavior. However, behavioral and neural evidence on the relationships between honesty and trust is missing. Here, combining a novel paradigm that successfully induces trustworthiness impressions with functional MRI and multivariate analyses, we demonstrate that honesty-based trustworthiness is represented in the posterior cingulate cortex, dorsolateral prefrontal cortex and intraparietal sulcus. Crucially, brain signals in these regions predict individual trust in a subsequent social interaction with the same partner. Honesty recruited the ventromedial prefrontal cortex (VMPFC), and stronger functional connectivity between the VMPFC and temporoparietal junction during honesty encoding was associated with higher trust in the subsequent interaction. These results suggest that honesty signals in the VMPFC are integrated into trustworthiness beliefs to inform present and future social behaviors. These findings improve our understanding of the neural representations of an individual's social character that guide behaviors during interpersonal interactions.

[1] Department of Psychology I, University of Lübeck, Lübeck, Germany. [2] Department of Education and Psychology, Freie Universität Berlin, Berlin, Germany. [3] Department of Decision Neuroscience and Nutrition, German Institute of Human Nutrition (DIfE), Nuthetal, Germany. [4] WZB Berlin Social Science Center, Berlin, Germany. [5] Charité-Universitätsmedizin Berlin, Corporate member of Freie Universität Berlin, Humboldt-Universität zu Berlin, and Berlin Institute of Health, Neuroscience Research Center, Berlin, Germany. [6] Deutsches Zentrum für Diabetes, Neuherberg, Germany. *email: gbellucc@gmail.com

Trust is the essential component of social life enabling successful cooperation and fostering individuals' well-being. The factors that induce trust in others remain, however, still largely unexplored. To date, at least two accounts have been proposed to explain an individual's trust.

One account proposes that interacting agents focus on maximizing their personal payoffs during social exchanges[1]. This account assumes that optimally rational agents trust another as long as they will be better off with trusting than distrusting[2]. Empirical investigations implementing economic games such as the trust game (TG) confirm that people are willing to trust as long as trusting leads to monetary rewards[3,4]. However, trust levels drop significantly when external incentives lack or when trust leads to monetary losses[5,6].

An alternative account argues that individuals take into account the social character and attitudes of the interacting partner when trusting. In this regard, individuals seek to form beliefs about the other's social character by focusing on whether the other's behavior fosters fairness, equality, and cooperation[7,8]. Honesty, that is, the quality of being reliable and the tendency to share truthful information, has been proposed as a central determinant of trustworthiness impressions promoting prosocial behaviors[9,10]. For instance, altruistic behavior, unconditional kindness, and reciprocity have been observed in response to others' honesty[11–14]. However, whether honesty also encourages others to trust is yet unexplored.

These two accounts make different predictions on the neural mechanisms underlying trust. When individuals focus on the trade-off between advantageous and disadvantageous consequences following a trust decision, brain regions signaling actual, or hypothetical decision outcomes (such as the ventral striatum and dorsal anterior insula) should be recruited in trusting interactions[15–18]. On the contrary, if trust draws on the social character of the other, brain regions associated with social evaluations (such as the ventromedial prefrontal cortex, VMPFC, and dorsolateral prefrontal cortex, DLPFC), and inferences on the other's intentions (e.g., the posterior temporoparietal junction, pTPJ) should be engaged during trusting behaviors[19–22]. However, to date, evidence on the brain regions representing the honest character of another is still missing.

In this study, we investigated for the first time whether information about the other's honest character evokes trustworthiness impressions that predict future trust in the other. Importantly, a reputation as a trustworthy person has been suggested to impact information processing during social learning. In particular, although individuals prefer to interact with, and learn from, trustworthy partners[23], beliefs about the other's trustworthiness bias how information from the trustworthy other is processed and learnt[24,25]. An explanatory hypothesis for such bias posits that beliefs about the other's trustworthiness modulate evaluations of information from trustworthy others. For instance, previous work has linked biased beliefs about others' reciprocity to differences in how information is encoded in the orbitofrontal cortex (OFC)[26], a region of pivotal importance in value representation[27]. However, it is still unknown whether a reputation as an honest person modulates information encoding and whether the OFC plays a role in such biased information processing.

Here, we developed a trust-inducing paradigm (take advice game, TAG), which enables us to isolate social evaluation signals related to the other person's trustworthiness (learnt through her honest and dishonest behavior) from nonsocial value signals related to one's task performance (neural responses to winnings and losses). Being able to disentangle these two types of information was of pivotal importance to the two main objectives of this study. On the one hand, it allowed us to isolate brain signals related to representations of the other's honest character. On the other, it enabled us to investigate any modulatory effects of the other's honest character on information processing. In the TAG, participants, in the role of advisee, had to learn the trustworthiness of advisers from feedback about their honest or dishonest advice. After the TAG, participants, now in the role of investor, played a one-shot TG with the advisers who advised them previously.

Using multivariate voxel pattern analysis (MVPA) in combination with functional magnetic resonance imaging (fMRI), we examined the relationships between honesty, dishonesty, and trust on the behavioral and neural level. On the behavioral level, honest behavior increases trust irrespective of proximal benefits associated with the act of trust. On the neural level, the honesty-based trustworthiness of the partner is represented in the posterior cingulate cortex (PCC), bilateral DLPFC, and left intra-parietal sulcus (IPS). Importantly, neural signal in these brain regions predicts an individual's willingness to trust the partner in a subsequent interaction. Further, enhanced integration of honesty into trustworthiness beliefs via stronger VMPFC-pTPJ connectivity is associated with higher trust levels later on. Finally, the partner's honest character modulates neural responses to positive and negative outcomes in the OFC.

## Results

**Paradigms.** In the TAG (Fig. 1a and Supplementary Fig. 1), participants in the role of advisee had to rely on the advice of different advisers to choose the highest of two cards. As participants did not have any information about the cards' numbers, they depended on the honesty of the advisers for their decisions. The advisers, on the other hand, could see only one of the two cards, that is, they knew more than the advisees, but did not have complete information about the cards. Hence, their advice was not about the winning card participants should pick, but rather additional information about the number of one of the two cards. In each trial, participants were paired with a different adviser (adviser phase). After the adviser sent his advice (advice phase), the advisee decided which card she wanted to pick (decision phase). Finally, the cards were disclosed to the advisee (feedback phase), who could see whether the adviser had been honest and whether she won or lost in that trial. Participants could win/lose €1 in each trial by choosing the card with the higher/lower number. After the TAG, participants in the role of investor played a one-shot TG with each of the advisers now in the role of trustee (Fig. 1b). Investors were paired with each trustee and received an initial endowment of 10 monetary units (MUs) that they could share with the partner. Investors were told that the shared amount would be tripled by the experimenter and passed on to the trustee who, in turn, could decide to share back any amount of it.

**Link between honesty and trusting behavior.** First, we tested whether honesty is associated with higher trust levels across contexts and regardless of proximal gains. In the TAG, individuals should be more willing to take the advice of honest advisers and distrust the advice of dishonest advisers. Our results demonstrate that participants took on average more advice from honest than dishonest others ($t_{(30)} = 3.68$; $p < 0.001$; 95% confidence interval (CI) = [0.03, 0.10]; Cohen's $d = 0.7$; Fig. 2a). Importantly, participants grounded their decisions to take an advice in the trustworthy character of the adviser (i.e., whether the adviser was honest or dishonest; $\beta = 0.38$; standard error (SE) = 0.12; 95% CI = [0.14, 0.62]; $p = 0.007$). On the contrary, monetary winnings and losses did not impact participants' decisions to take an adviser's advice ($\beta = -0.001$; SE = 0.07; 95% CI = [−0.14, 0.14]; $p = 0.980$; Table 1). This suggests that our participants trusted an adviser based on the adviser's trustworthy

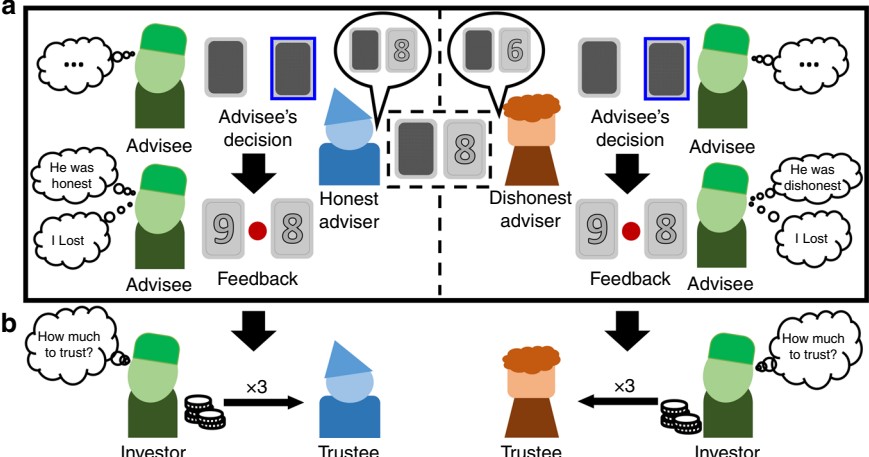

**Fig. 1** Paradigms. **a** Schematic representation of the take advice game (TAG). Advisers were given information about one of the two cards and could communicate this information to the advisee. Participants, in the role of advisee, made a decision based on the information received (decision phase). In the feedback phase, advisees saw the actual numbers on the cards, which informed them about the adviser's honest behavior (honest vs. dishonest), and a green or red circle, which informed them whether they won or lost, respectively. **b** After the TAG, participants in the role of investor played a one-shot trust game (TG) with the advisers now in the role of trustee. Investors received a monetary endowment and decided whether they wanted to entrust some of this amount with the trustees. Investors were told that the shared amount was tripled by the experimenter and passed on to the trustee, who could decide to share back any portion of the tripled amount. See also Supplementary Fig. 1

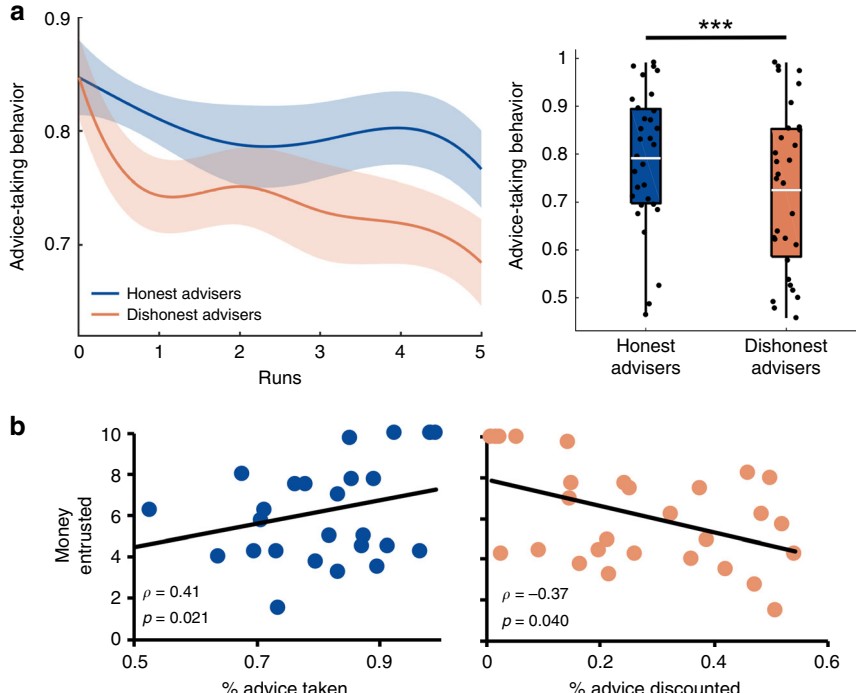

**Fig. 2** Behavioral results. **a** Trusting behavior in the take advice game over runs (left) and on average (right) toward honest and dishonest advisers. On average, participants took significantly more advice from the honest than the dishonest adviser (*t* test). Data points on the left were interpolated for visualization purposes and shadowed areas represent standard errors. White lines in the box-plots on the right represent average advice-taking behavior across participants. Each black dot represents one participant. **b** Amount of money entrusted in the trust game with honest (left) and dishonest (right) others correlated with participants' willingness to take advice from the advisers (Spearman's correlations). Each dot represents one participant. ***P < 0.001

behavior and irrespective of their proximal benefits. Indeed, the majority of our participants ($M = 88.2\%$) explicitly reported in an exit questionnaire (see Methods) that their decisions were based on the trustworthiness and advice of the advisers. Importantly, participants applied such trustworthiness-based strategy even though they were aware that it was not successful to gain more benefits ($\chi^2 = 13.68$, $p = 0.0002$).

Moreover, although trust in the other's advice was comparable for both honest and dishonest advisers in the very first trials of the TAG, participants quickly adjusted their behavior to the other's honesty over the course of the social interaction (Fig. 2a). Indeed, participants' advice-taking behaviors toward the two advisers differed increasingly over time ($\beta = 0.01$; SE $= 0.006$; 95% CI $= [0.0001, 0.024]$; $p = 0.048$), especially due to a

**Table 1 Mixed-effects logistic regression analysis of advice-taking behavior**

| Regressor | $\beta$ (SE) | CI |
|---|---|---|
| Intercept | 2.01 (0.33)** | 1.37, 2.64 |
| Honest adviser | 0.38 (0.12)* | 0.14, 0.62 |
| Advised number | −0.17 (0.07) | −0.30, −0.03 |
| Advised card | 0.02 (0.07) | −0.11, 0.15 |
| Feedback previous trial | −0.001 (0.07) | −0.14, 0.14 |

$\beta$ coefficients (standard errors) from the generalized mixed-effects logistic regression model with maximal random-effects structure predicting advice-taking behavior (1 = advice taken; 0 = advice not taken). $P$ values were based on a likelihood ratio test
SE standard error, CI confidence interval
*$p < 0.01$; **$p < 0.001$

significant, linear decrease in trust in the advice of dishonest advisers ($\beta = -0.02$; SE $= 0.007$; 95% CI $= [-0.028, -0.002]$; $p = 0.021$). On the contrary, advice-taking behavior toward honest advisers did not significantly change over time ($\beta = -0.005$; SE $= 0.006$; 95% CI $= [-0.016, 0.007]$; $p = 0.410$).

Second, we investigated whether these specific effects of the other's trustworthiness on advice-taking behavior in the TAG generalize to a different context and measure of trust (i.e., the TG). Our results confirm this, showing that advice-taking behavior in the TAG correlated with subsequent, economic trust decisions in the TG on average ($\rho_{(29)} = 0.39$; $p = 0.031$), and separately for both honest ($\rho_{(29)} = 0.41$; $p = 0.021$) and dishonest advisers ($\rho_{(29)} = -0.37$; $p = 0.040$). That is, the more likely participants were to trust the advice of an adviser, the more willing they were to entrust that adviser with money in a subsequent interaction (Fig. 2b). As expected, the amount of money shared with the advisers in the TG did not significantly correlate with participants' monetary winnings in the TAG either on average ($\rho_{(29)} = 0.17$; $p = 0.350$) or separately for the two advisers (honest adviser: $\rho_{(29)} = 0.30$; $p = 0.106$; dishonest adviser: $\rho_{(29)} = 0.01$; $p = 0.978$). These results confirm that economic trust decisions in the TG did not represent a form of repayment for the benefits participants obtained from the adviser's advice in the previous interaction, but rather reflected participants' willingness to trust the adviser's honesty in advice giving.

Finally, we checked the proportion of positive and negative feedback received by our participants. Participants received on average the same amount of positive and negative feedback (mean difference $= 0.0013 \pm$ SD $= 0.07$; $t_{(30)} = 0.11$; $p = 0.916$), despite more positive feedback when interacting with honest than dishonest advisers (honest advisers: $M = 63.5\% \pm$ SD $= 7.4$; dishonest advisers: $M = 56.7\% \pm$ SD $= 5.0$; $t_{(30)} = 4.09$; $p < 0.001$).

**Neural representations of trustworthiness**. Next, we examined the neural patterns of the advisers' trustworthy character inferred from their honest behavior and value information related to participants' performance. In doing so, we investigated whether these neural patterns capitalize on similar brain regions informative of individual trust. Our task design elegantly allows this, since in the feedback phase, participants received information about both the other's trustworthiness (honest/dishonest behavior) and their own task performance (winnings/losses). Hence, by applying a whole-brain searchlight MVPA to neural activations during the feedback phase with a leave-one-run-out cross-validation (LOROCV) procedure (Fig. 3a), we could separately decode trustworthiness and value information to identify brain regions belonging to a trustworthiness decoding network and value decoding network, respectively. To this end, a support vector machine (SVM) was trained on $\beta$ parameters estimated using two general linear models

(GLMs) that coded trustworthiness information (GLM1) and value information (GLM2) in the feedback phase (see Methods).

The trustworthiness decoding network revealed clusters with classification accuracy above chance in the PCC, right and left DLPFC, and left IPS (cluster-level, family-wise error corrected, FWEc, <0.05; Fig. 3b and Supplementary Table 1). Signal in these brain regions was able to classify the neural signatures of honesty and dishonesty of out-of-sample individuals with 68% accuracy (sensitivity: 68%; specificity: 68%; $p < 0.0001$, based on a nonparametric test of 10,000 permutations; Fig. 3c). On the contrary, the value decoding network consisted mainly of regions in the medial PFC extending from the anterior cingulate cortex (ACC) to the striatum (voxel-level FWE <0.05; Fig. 3d and Supplementary Table 1). Signal in these brain regions was able to classify the neural signatures of positive and negative outcomes of out-of-sample individuals with 82% accuracy (sensitivity: 87%; specificity: 77%; $p < 0.0002$; Fig. 3e). Hence, these analyses indicate a specific neural network representing the other's social character (i.e., trustworthiness) that could be separated from neural signal representing value information. To note, classification accuracy of value information was much better than classification accuracy of social character information. These results concur with previous findings[28] and may hinge on the nature of social concepts, which are distributed neural representations that might be difficult to fully capture using an anatomical-based searchlight approach.

Finally, we set out to characterize the peculiar functional associations of the trustworthiness decoding network. We first ran GLM analyses to control for possible confounds of the observed neural patterns. In particular, we computed another GLM1 adding parametric modulators to the feedback phase for risk (as mean-squared deviation from the expected outcome given the adviser's advice) and congruency (as deviance of the adviser's advice from the actual card number on the advised card). These analyses revealed that our results hold also after controlling for these potential confounding factors (Supplementary Fig. 2).

Second, using meta-analytic functional decoding (neurosynth.org)[29], we quantitatively evaluated the representational similarity of the trustworthiness decoding network with neural activation patterns associated with specific psychological components. In particular, we compared the neural signatures of trustworthiness in our study against reverse inference meta-analytic neural patterns of neural images of previous studies stored in the Neurosynth database and associated with particular psychological terms. For this analysis, we chose twelve terms associated with the social and nonsocial domains, such as social cognition, theory of mind, rewards, congruency and risk (Supplementary Fig. 3). Results demonstrate that the trustworthiness decoding network was preferentially associated with psychological terms related to mentalizing and social cognition (Supplementary Fig. 3), validating the ability of our task in singling out neural patterns that likely underlie the formation of trustworthiness beliefs about the advisers. Next, we set up to test this peculiar functional role of the trustworthiness decoding network in representing the trustworthy character of others.

**Neural representations of trustworthiness predict trust**. A central feature of the neural representation of a character trait, such as trustworthiness, is its ability to inform decisions across contexts[30]. Thus, neural patterns decoding the other's trustworthiness (i.e., within the trustworthiness decoding network, but not within the value decoding network) should be able to predict individual trust decisions in the TG. To test this, a multivariate prediction analysis with a LOSOCV procedure was performed. Prediction significance was tested against a random distribution

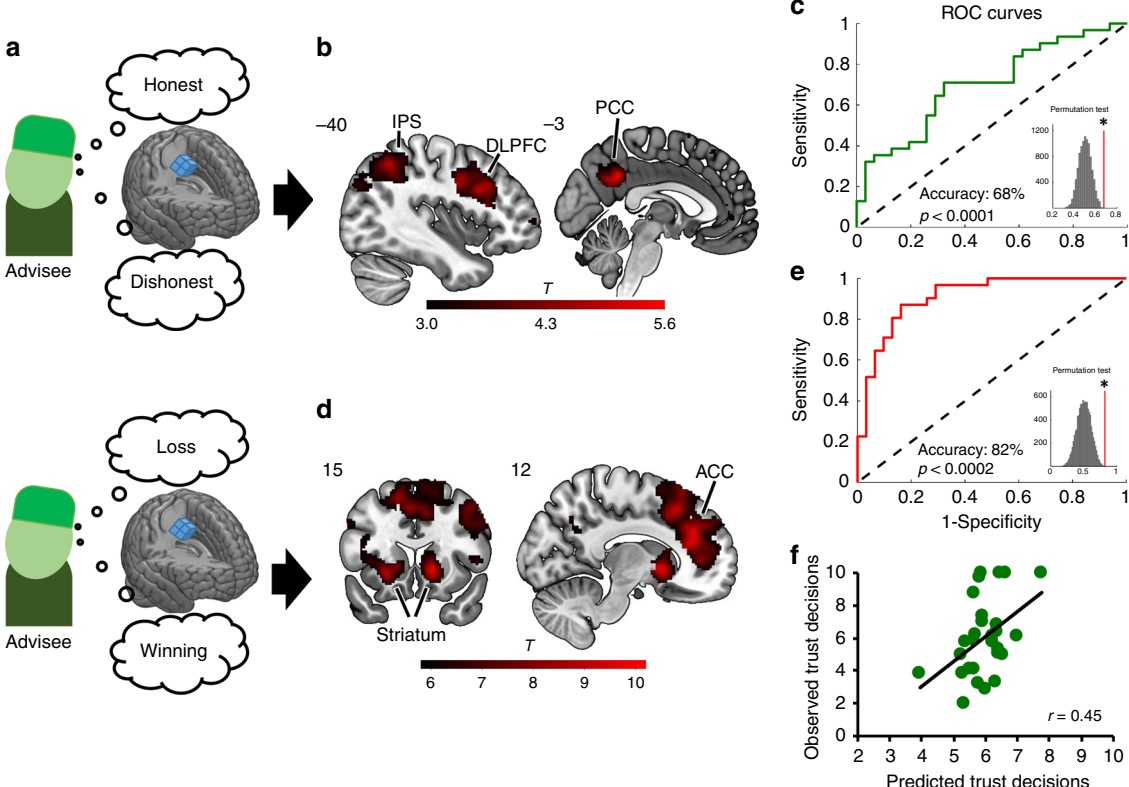

**Fig. 3** Decoding honesty and predicting trust. In two MVPAs applied to the feedback phase of the TAG (**a**), a support vector machine (SVM) was trained to decode honest and dishonest advice (GLM1) to determine the trustworthiness decoding network (upper), and to decode winnings and losses (GLM2) to determine the value decoding network (lower). The trustworthiness decoding network (**b**) included brain regions such as the PCC, DLPFC, and IPS, and could successfully distinguish neural signatures of honesty and dishonesty in out-of-sample individuals (**c**). The value decoding network (**d**) included the striatum and ACC and could successfully distinguish neural signatures of winnings and losses in out-of-sample individuals (**e**). Finally, a multivariate prediction analysis with support vector regression (SVR) showed that the neural patterns of the trustworthiness decoding network successfully predicted individual economic trust decisions in the TG, thereby showing across-context generalizability (**f**). Both out-of-sample classification and prediction analyses were based on a leave-one-subject-out cross-validation procedure and their significance tested using a permutation test with 10,000 permutations. Each dot represents one participant. See also Supplementary Fig. 2 and Supplementary Table 1. MVPA, multivariate voxel pattern analysis; TAG, take advice game; PCC, posterior cingulate cortex; IPS, intraparietal sulcus; DLPFC, dorsolateral prefrontal cortex; ACC, anterior cingulate cortex; TG, trust game. Heatmap represents *t* values

of 10,000 permutations. Results demonstrate that the trustworthiness decoding network significantly predicted the amount of money entrusted in the TG by out-of-sample individuals (standardized mean-squared error, smse, = 0.80; $p < 0.007$; Fig. 3f and Supplementary Fig. 4). On the contrary, the predictive model using the neural signal of the value decoding network did not yield a significant prediction (smse = 1.06; $p = 0.84$; Supplementary Fig. 4). By showing that neural patterns decoding trustworthiness information about others predict an individual's willingness to trust in a different social context, these findings indicate a peculiar functional role of those trustworthiness-decoding brain regions in representing behaviorally relevant information about another person's social character.

**Stronger integration of honesty signals correlates with higher trust.** MVPA identified neural patterns of brain signal entailing information about another person's trustworthiness that were informative of an individual's trusting behavior and were different from neural patterns related to value information. To further characterize brain regions more strongly recruited by honesty and dishonesty, and to test whether and how honesty modulates neural responses to value information, whole-brain univariate analyses were performed on the brain signal during the feedback phase.

Contrast analyses between honesty and dishonesty revealed that dishonesty more strongly activated bilateral DLPFC, left IPS and IPL (FWEc <0.05; Fig. 4a and Supplementary Table 2), while the VMPFC and ACC were significantly more engaged by honesty (FWEc <0.05; Fig. 4b and Supplementary Table 2). These results indicate a stronger reliance of dishonesty on brain regions within the trustworthiness decoding network, suggesting that dishonesty likely requires recruitment of brain regions representing the other's character to constantly optimize one's beliefs about the other. On the contrary, honesty more strongly relied on medial prefrontal areas associated with evaluations of positive qualities of others and self.

The definition of two separate GLMs (i.e., GLM1 and GLM2) was necessary to estimate separate β images to train the machine learning algorithms in our previous multivariate classification and regression analyses. However, this leaves the question unanswered as to whether the observed neural signatures for honesty and dishonesty are specific to social information processing or are confounded by neural signatures of nonsocial value information processing. We thus tested the specificity of our findings by comparing the univariate results yielded by the two separate GLMs with results of a single parametric GLM, including only one feedback regressor and two categorical, parametric modulators (one coding for value information and one for the

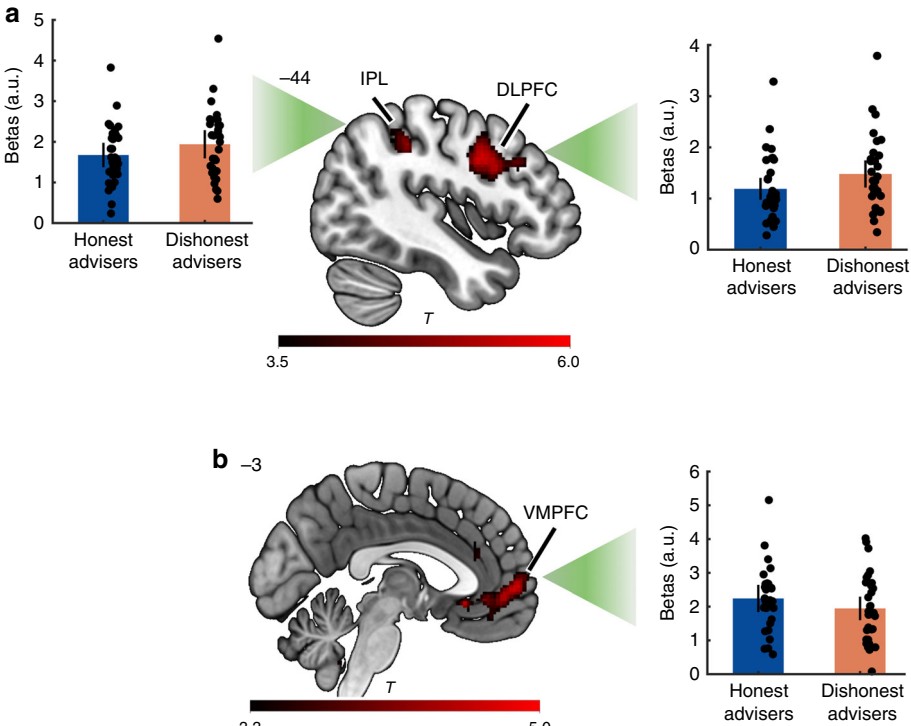

**Fig. 4** Honesty vs. Dishonesty. Univariate contrasts revealed that brain areas within the trustworthiness decoding network (i.e., IPL and DLPFC) were more engaged by dishonesty than honesty (**a**), whereas honesty more strongly recruited the VMPFC (**b**). Error bars indicate standard errors across participants. Each dot represents one participant. See also Supplementary Table 2. IPL, inferior parietal lobule; DLPFC, dorsolateral prefrontal cortex; VMPFC, ventromedial prefrontal cortex; a.u., arbitrary units. Heatmap represents *t* values

adviser's trustworthiness; see Methods). Notably, this single parametric GLM allowed us to control for spurious signal by orthogonalizing the two parametric modulators. As can been seen in Supplementary Fig. 5, our results hold also with this GLM definition, suggesting that the observed neural signatures of trustworthiness are specific to social information processing.

Next, as the VMPFC has previously been shown to be functionally connected with brain regions associated with social cognition during socially relevant computations[31], we reasoned that honesty signals in the VMPFC may be integrated into beliefs about the other's social character via functional connectivity with brain regions associated with social cognition. To define these potential pathways, a task-dependent functional connectivity analysis was implemented using the VMPFC as seed region. This functional connectivity analysis shows that the VMPFC was more strongly coupled to the left pTPJ (−40, −50, 30, *x*, *y*, *z*; FWEsvc <0.05; Fig. 5a) during honesty encoding than dishonesty encoding. We then reasoned that if the information flow between the VMPFC and left pTPJ during the feedback phase were specifically associated with the formation of trustworthiness beliefs about another person, the strength of this connectivity should be related to subsequent trust decisions, but not to individual monetary winnings. Indeed, functional connectivity between the VMPFC and left pTPJ during honesty and dishonesty encoding in the TAG significantly correlated with the amount of money entrusted in the TG to honest ($\rho_{(29)} = 0.54$; $p < 0.002$) and dishonest ($\rho_{(29)} = 0.48$; $p = 0.006$) advisers (Fig. 5b). On the contrary, no significant correlations were found between individual winnings and the VMPFC-pTPJ connectivity for either honest ($\rho_{(29)} = 0.29$; $p = 0.111$) or dishonest ($\rho_{(29)} = 0.11$; $p = 0.542$) advisers (Fig. 5c). These results suggest that functional connectivity between the VMPFC and left pTPJ likely reflects integration of honesty information into knowledge about

the other's social character. Specifically, stronger integration of honesty signal from the VMPFC into the pTPJ led to higher trust in the adviser during a subsequent interaction, reflecting our behavioral findings that the advisers were trusted more later on the more participants believed them to be honest.

**Honesty biases value information processing**. We then turned to test whether and how these specific activation patterns of honesty and dishonesty modulate brain responses to value information during the feedback phase. Previous behavioral studies have suggested that positive qualities of others bias information processing[24,25]. Such a bias may hinge on trait-dependent differences in neural responses to novel information. We tested this hypothesis by looking at how honesty and dishonesty modulate neural responses to positive and negative outcomes (i.e., GLM3, see Methods).

We first examined the neural responses to positive and negative outcomes during interactions with honest and dishonest advisers separately. Positive outcomes during both interactions with honest and dishonest advisers elicited similar activations in the striatum, and for honest advisers, these activations extended to the OFC (Supplementary Table 3). Similarly, negative outcomes engaged the middle cingulate cortex and inferior frontal gyrus for both honest and dishonest advisers (Supplementary Table 4). Next, we investigated the modulatory effects of honesty and dishonesty on positive and negative outcomes. This analysis revealed that brain regions encoding positive and negative outcomes were differently modulated by the honest character of the advisers. In particular, neural signal in the parietal cortex was modulated by dishonesty during both positive (right IPL; FWEc <0.05; Fig. 6a) and negative (left IPS; FWEc <0.05; Fig. 6b) outcomes (Supplementary Table 5). On the contrary,

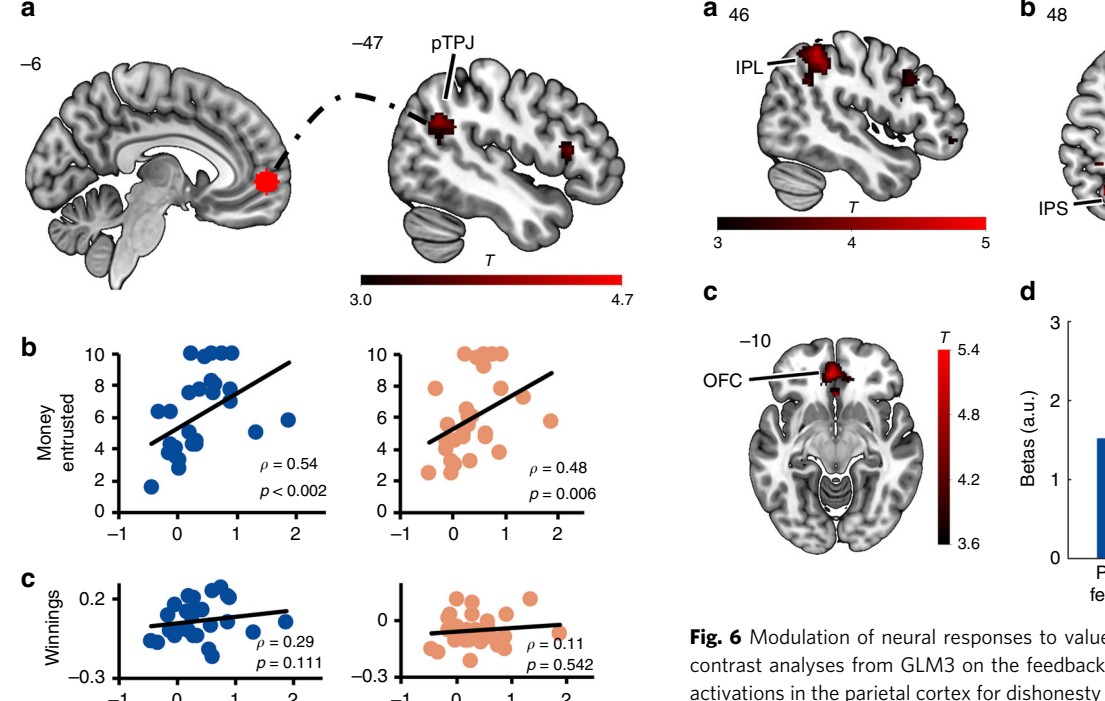

**Fig. 5** Task-based functional connectivity analysis. Task-based functional connectivity between the VMPFC and left pTPJ was stronger for honesty than dishonesty (**a**). Critically, this functional connectivity correlated with an individual's willingness to trust in the TG (**b**), but not with one's payoffs in the TAG (**c**) (Spearman's correlations). Blue dots on correlation plots on the left represent behaviors toward honest advisers, orange dots on correlation plots on the right represent behaviors toward dishonest advisers. Each dot represents one participant. VMPFC, ventromedial prefrontal cortex; pTPJ, posterior temporo-parietal junction; PPI, psychophysiological interaction; a.u., arbitrary units. Heatmap represents $t$ values

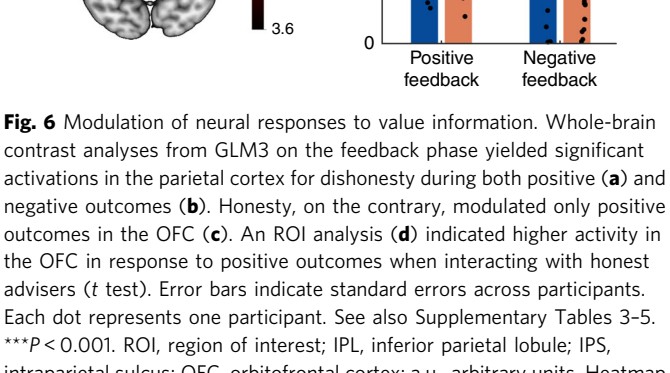

**Fig. 6** Modulation of neural responses to value information. Whole-brain contrast analyses from GLM3 on the feedback phase yielded significant activations in the parietal cortex for dishonesty during both positive (**a**) and negative outcomes (**b**). Honesty, on the contrary, modulated only positive outcomes in the OFC (**c**). An ROI analysis (**d**) indicated higher activity in the OFC in response to positive outcomes when interacting with honest advisers ($t$ test). Error bars indicate standard errors across participants. Each dot represents one participant. See also Supplementary Tables 3–5. ***$P < 0.001$. ROI, region of interest; IPL, inferior parietal lobule; IPS, intraparietal sulcus; OFC, orbitofrontal cortex; a.u., arbitrary units. Heatmap represents $t$ values

modulation of neural responses to outcomes by honesty was found only in the OFC during positive outcomes (FWEc <0.05; Fig. 6c). These results indicate an asymmetry in the neural responses to positive and negative outcomes for honesty.

Using an independent region of interest (ROI) in the OFC, we more closely examined in a post-hoc ROI analysis this asymmetric modulation of positive outcomes by honesty (Fig. 6d). Activity in the OFC was significantly higher in response to positive outcomes when interacting with honest advisers as opposed to dishonest advisers (honesty: $M = -0.08$; SD = 0.46; dishonesty: $M = -0.44$; SD = 0.35; $t_{(30)} = 4.72$; $p < 0.0001$, CI = [0.21, 0.52]; Cohen's $d = 0.85$), while OFC activity during negative outcomes was comparable for the two advisers (honesty: $M = -0.45$; SD = 0.71; dishonesty: $M = -0.57$; SD = 0.65; $t_{(30)} = 1.16$; $p = 0.257$, CI = [−0.10, 0.36]; Cohen's $d = 0.21$). This asymmetry in the neural responses to positive outcomes in the OFC suggests that value information processing may be biased during interactions with honest individuals.

## Discussion

Understanding others is pivotal for successful cooperation. In particular, other people's character may function as a proxy for their likely behavior in a future encounter. Thus, trustworthy partners are likely to be trusted in the future, while untrustworthy others are likely to be avoided. In this study, we showed that the honest character of an adviser makes people more likely to accept

the adviser's advice and more willing to trust the adviser in a subsequent interaction. Moreover, neural signatures of the partner's trustworthiness in the DLPFC, IPS, and PCC predicted individual willingness to trust the partner later on, and stronger integration of an honesty signal from the VMPFC into the pTPJ correlated with higher future trust in the partner.

When no prior information about how a partner will behave is provided, individuals try to gather evidence about the partner's social character to inform their decisions about what to do when interacting with that partner. Over the course of multiple interactions, information about the partner's current behavior lays the groundwork for the formation of beliefs about the other's reputation[32]. Consistently with previous models of trust[9], being reliable and telling the truth contributes to an honest reputation that made participants more likely to accept advice. On the contrary, when participants realized that their initial trust in the partner's advice was misplaced, they increasingly discounted the advice of dishonest advisers. Interestingly, even though the adviser's advice was not associated with the best option in the game and did not bring higher benefits to the participants, participants repaid the advisers for their honesty in advice giving during a future trusting interaction.

As there were no incentives for the advisers to help the advisees (except goodwill or a good reputation) and the advisees did not commit to reciprocate, the dynamics in play in our study resemble real-life scenarios where individuals need to interact with each other without requirements or guarantees from the interacting partner. For instance, trusting someone to give good advice or keep a secret is an act of trust triggered by impressions of the partner's trustworthiness without the requirement of an initial generous act by the partner[33]. In these contexts, individuals

likely assume that the other person would comply with the shared social norms, which represent a cluster of expectations an individual can use to make good-enough estimations of another person's behavior[34]. Over the course of multiple interactions, individuals need to quickly infer the trustworthiness of the other person on the basis of what they have learned from their actual behavior and might eventually consider adopting better behavioral strategies for current and future encounters with that person[35].

Thus, trusting someone else in a social interaction requires the ability to form a belief about the other's character (i.e., who the other as a person is) and tailor one's behavior to the other's actions and intentions. In our study, we observed that the partner's trustworthiness (inferred from her honest or dishonest behavior) was decoded in four brain regions (i.e., the PCC, left IPS, and bilateral DLPFC), which were able to successfully classify neural responses to honesty and dishonesty in out-of-sample individuals. In particular, recruitment of the PCC, a central hub of the mentalizing brain system[36,37], is likely related to cognitive processes associated with trait judgments[38], while the IPS, in line with its role in processing expectations related to current goals and stimulus–response selection[39], likely sustains attribution of temporary beliefs to tune action selection[38,40]. Finally, the DLPFC might be responsible for translating the knowledge about the partner into action. In particular, in line with its role in generous decisions[41] and group-based cooperation[42,43], the DLPFC might be involved in the decision to engage in prosocial actions in response to the partner's behavior.

Crucially, these brain regions have previously been observed to be interconnected during interpersonal interactions. In particular, the left IPS shows selective connectivity with the DLPFC and PCC while understanding others during social interactions[30,44,45], suggesting that these brain regions build an intertwined brain network engaged in representations of socially-relevant qualities of others. These representations may underlie individual, behavioral attitudes based on which adequate behaviors tailored to the current partner's character and reputation are flexibly adopted. Moreover, such representations might be retrieved in future interactions with the partner, as their content is informative of the partner's character and is thus useful to sustain individual choices that strongly rely on those character impressions. In line with this, in our study, we observed that neural signal in the PCC, left IPS, and bilateral DLPFC predicted the future willingness to trust the partner in a social context (i.e., in the TG), where participants made trust decisions based exclusively on their trustworthiness impressions of the partner from the previous interaction (i.e., from the TAG).

Critically, the IPS and DLPFC, together with the IPL, were also more strongly recruited by dishonesty as opposed to honesty. These findings are in line with previous evidence that the IPS is consistently activated by others' non-cooperative behavior[46], and that the DLPFC, together with the IPL, tracks violations of expectations[47,48] and decisions to lie[49]. The stronger recruitment of these brain regions by dishonesty might reflect the need to constantly track the behaviors of dishonest partners for an online belief updating and a flexible behavior revision. In fact, on the behavioral level, we observed that advice-taking behavior toward honest advisers did not significantly change over time, whereas participants continuously adjusted their advice-taking behavior for dishonest advisers with a consistent decrease of trust in them over time. These results suggest that the recruitment of the DLPFC, IPS, and IPL is more strongly required in cases of norm-deviant behaviors (e.g., when other people are dishonest, unfair, or noncooperative) to carefully track the other's actions and optimally adjust one's own behavior.

On the contrary, honesty was observed to more strongly recruit the VMPFC, a brain region previously associated with behaviorally-relevant representations of positive traits of others[30,50,51]. In particular, the VMPFC was functionally coupled to the left pTPJ during honesty encoding in the TAG, and the strength of this functional connectivity was further correlated with higher trust in the adviser during the later interaction in the TG. In line with the pTPJ role in processing inferences on others' mental states[52,53] and social prediction errors[54,55], these findings suggest that inferences on the partner's intentions undertaken by the pTPJ might be supported by integration of novel, incoming information about the partner's honesty encoded in the VMPFC. Interestingly, a recent work has indicated that connectivity of the left pTPJ with other social cognition regions supports behavioral trust and that the experimentally-induced disruption of trust (via aversive affect) was concomitantly followed by the suppression of pTPJ connectivity during trust decisions. These findings suggest a pivotal role of pTPJ connectivity in integration of behaviorally-relevant signal[56]. In our experiment, stronger integration of an honesty signal likely led to more positive beliefs about the partner's intentions, increasing one's willingness to trust. Thus, the interplay between the VMPFC and left pTPJ represents a central neural mechanism underlying integration of character information for behaviorally-relevant inferences on others' actions and intentions.

Finally, we observed modulation of neural responses to value information by honesty in the OFC during outcome evaluations. Specifically, higher OFC activity was observed for positive outcomes received when interacting with honest partners. These results suggest that in line with its role in processing subjective values of both social and nonsocial rewards[57,58], higher neural activity in the OFC reflects an enhanced subjective value of nonsocial rewards induced by the positive character of the interacting partner. These neural findings might provide a mechanistic explanation for the positivity bias toward individuals with a good reputation that has been observed to influence learning processes[24,25]. Given the OFC role in learning mechanisms[59,60], an asymmetry in the representation of positive and negative events associated with an individual of good social qualities in the OFC might promote a stronger susceptibility to reputational priors and a reduced flexibility in the revision of one's beliefs about the partner. Consistently with that, decreased OFC activity has previously been associated with stronger resistance to political belief change during information encoding[61]. Hence, an asymmetric valuation of new incoming information likely contributes to judgmental biases and suboptimal learning.

Taken together, our results improve our understanding of how neural patterns representing honesty-based trustworthiness guide social behaviors in interpersonal interactions. The PCC and frontoparietal brain regions represent behaviorally-relevant knowledge about the other's social character, likely taking a role in the flexible revision of one's current behavior for optimal adaptation to the partner's actions. Further, social behaviors such as trust are likely enacted based on the integration of character information from the VMPFC into the left pTPJ for reliable inferences on the good intentions of the partner. Finally, an asymmetric activity in the OFC in response to positive outcomes due to the good reputation of the interacting partner likely jeopardizes an individual's ability to optimally form and update one's beliefs about the other, fostering a broad array of judgmental biases.

Although we here showed that trustworthiness-related neural signal successfully predicts individual trust decisions in a future social interaction, future studies are still needed to investigate in a brain-to-brain predictive framework whether these neural signatures of trustworthiness are also able to predict the neural

correlates of individual trust decisions. Further, future studies, especially in advice-taking paradigms, might also consider controlling for individual susceptibility to social influence, which might, for instance, explain an individual's propensity to take advice from others. Another interesting research question for future studies relates to how other factors of trustworthiness impressions (like competence and benevolence) interact with honesty to elicit trust and/or distrust in others. By shedding light on how social characters are represented in the brain and influence individual decisions, this work makes an important contribution to the extant literature on human cognition in a broad range of scientific fields, such as neuroscience, social psychology, sociology, economics, and political sciences.

## Methods

**Subjects**. Thirty-one participants (20 females) participated in the experiment (age: $24.29 \pm 3.81$ $M \pm SD$). Participants were recruited from the student community at the University. They were all right handed and had no history of neurological or psychiatric disorders. Participants gave written informed consent after a complete description of the study was provided. All the procedures involved were in accordance with the Declaration of Helsinki and approved by the Ethical Committee of the University of Lübeck, Lübeck, Germany.

**Take advice game**. In the TAG, participants played as advisee a card game with eight different advisers in a randomized order. Participants were told that these advisers were other participants who were taking part in the same experiment and were preparing themselves in other rooms. Participants were told that roles in the game were randomly assigned by drawing a ball with their role from a lottery box and that all participants were going to do it prior to the experiment. They were told that for transparency reasons, the ball-drawing procedure was going to be performed in front of a camera on top of a screen where each participant could see each of the participants in the other rooms drawing their role. However, to guarantee anonymity, all cameras were mounted on top of the screen so that each participant was recorded only up to the chin. Camera adjustments were performed prior to the ball-drawing procedure to assure this. Moreover, to further guarantee anonymity, each participant needed to choose an avatar that represented themselves in the game (Supplementary Fig. 1). In reality, participants received always the advisee role and the other videos were pre-recorded.

As advisee, participants' task was to draw the card with the higher number. Numbers on the cards ranged from 1 to 9 (except for 5). As participants did not have any information about the card numbers, they needed to rely exclusively on the adviser's advice for their decisions (establishing an adviser–advisee interdependency necessary for trust). Participants were told that the advisers could see only one of the two cards (adviser phase: 2–3 s) and could communicate this information to them (advice phase: 1 s). This implies that although advisers had more information than our participants, they did not know which card was the winning one, making this setting similar to real-life scenarios in which people generally ask for advice those who may know better, but advisers rarely have complete knowledge of life situations. Participants also knew that advisers could help them but did not benefit in doing so. However, both partners knew that after the TAG they were going to play a second game (i.e., the TG, see below), in which participants could repay the advisers for their honesty in advice giving. Thus, in the TAG, advisers were motivated to form a good reputation in the hope that participants would repay them later on. To note, however, participants did not promise or commit to repay the advisers for their advice. The dynamics set in motion by this design resembles real-life interactions in which honest behavior (e.g., giving good advice) has often no proximal benefits to an individual but may help her form a good reputation that might turn out advantageous in the future (a possible, distal benefit).

Moreover, to disentangle trustworthiness information about the advisers (honesty/dishonesty) from value information about participants' decisions (winnings/losses), the advice of honest advisers was made unpredictive of the winning card (i.e., 50% of the time information about the losing card was provided by the honest adviser). Thus, cards were drawn from a uniform distribution with pseudo-random sampling without replacement. The pseudo-random sampling procedure was optimized to have a realized probability of card drawing that approximates chance in both conditions, as would be expected in random drawing. A two-sample Kolmogorov–Smirnov test confirmed that the realized distributions of card numbers did not differ between advisers (K–S test = 0.25; $p = 0.929$). Participants then chose one of the two cards (decision phase: 1 s) and saw a final feedback (feedback phase: 1 s) in which they received both social information (the card numbers based on which they could infer the adviser's trustworthiness) and nonsocial information (a green or red circle representing winnings and losses, respectively). In each trial, participants could win or lose €1. Intertrial stimulus intervals were 2–8 s ($M = 2.6$ s) long, whereas jitters between trials were 2–8 s ($M = 4$ s) long. Participants played a total of 5 runs with 48 trials each (24 with honest and 24 with dishonest advisers) for a total of 240 trials.

Advice-taking behavior in the TAG was operationalized as the probability of choosing a card given the informativeness of the advice received. The optimal strategy in the game would be to choose more frequently a card when the adviser communicated that a number bigger than five was on that card but choose the other card when the adviser communicated that a number smaller than five was on that card. Moreover, as we manipulated the advisers' honesty with four honest advisers sending accurate information and four dishonest advisers sending inaccurate information (with 100% contingency), we hypothesized that participants would employ the optimal card-choice strategy differently for honest and dishonest advisers. Analyses of card choice probabilities confirmed our hypotheses (Supplementary Fig. 6). A repeated-measures analysis of variance with card numbers as repeated measure yielded a significant main effect of card number ($F_{(7,210)} = 83.13$; $p < 0.0001$; $\eta_p^2 = 0.74$) with participants being more likely to choose a card when a number higher than five was said to be on the card and less likely to do so otherwise. Importantly, an interaction effect between card number and advisers was also found ($F_{(7,210)} = 4.86$; $p < 0.0001$; $\eta_p^2 = 0.14$), suggesting that participants were more consistently employing the optimal strategy when interacting with honest advisers but not when interacting with dishonest ones.

To test the hypothesis that this interaction effect was due to the difference in trust in the advisers and was not simply driven by differences between specific cards, we ran post-hoc $t$ tests and compared the average choice probability for the honest and dishonest advisers for cards 1–4 and cards 6–9. Results indicate participants were less likely to choose a card when honest advisers told them a low number was on the card (honest vs. dishonest advisers for cards 1–4: $t_{(30)} = -2.97$; $p < 0.006$), but more likely to choose a card when honest advisers told them a high number was on the card (honest vs. dishonest advisers for cards 6–9: $t_{(30)} = 2.88$; $p = 0.007$). These results suggest that participants were discounting the advice of a dishonest adviser, likely because they did not believe it to be informative. In other words, this decrease in the likelihood of the use of the optimal strategy for dishonest advisers suggests a devaluation of their advice. Overall, these findings indicate that for the same piece of advice, the likelihood someone is going to take that advice hinges on their trust in the adviser or, complementary, on how much they value the adviser's advice (i.e., recognize it as informative).

Finally, it has to be noted that although the reputation dynamics set in motion by our design closely resemble real-life scenarios, the fact that advisers provided either correct or incorrect advice in every trial might not seem very realistic. As we had eight different advisers (four honest and four dishonest), the task was still challenging enough to look like completely unrealistic and participants needed to learn trial by trial the behavioral conduct of each adviser, as they were not instructed about the underlying behavioral contingency. We preferred a 100% contingency for the advisers' advice-giving behaviors to, for example, a probabilistic instantiation thereof because we wanted to avoid that the multivariate algorithm for the decoding of neural signal related to the social character of the advisers might have picked some spurious signal evoked, for instance, by the partner's behavioral inconsistency or prediction error associated with subjective expectancies. However, future studies might want to consider introducing some variability in the behavior of pre-programmed human-like agents.

**Trust game**. After the scanning session, participants played as investor a one-shot TG with the same partners who advised them in the TAG. Participants were endowed with 10 MUs for each adviser in the role of trustee and decided whether they wanted to share any of this initial endowment with them (economic trust decision). They were told that any amount they decided to share would be tripled by the experimenter and passed on to the trustee who could in turn decide to share back any portion of this tripled amount (reciprocity decision). The TG was used to probe the transfer effect of the honest reputation established in the TAG on individual trust in a new social interaction.

**Exit questionnaire**. To acquire an explicit measure of the criteria and motives behind participants' behavior in the TAG, after the experiment, participants were asked to report whether they used any particular strategy and whether they thought this strategy was successful (binary answer option). Although a significant portion of participants reported that they used a strategy in the TAG ($\chi^2 = 5.89$; $p = 0.015$), except for four participants, no one believed it was successful ($\chi^2 = 13.68$; $p = 0.0002$).

Moreover, they were also asked to describe the criteria for their decisions in the TAG (answering the question: "which strategy did you use for your choices in the first game?"). Three researchers blind to the study design and purposes categorized participants' free answers. The first rater identified three main strategies. The second and third raters identified further subcategories for a total of seven and eight categories, respectively. These could be grouped into the three main strategies of the first rater (averaged inter-rater reliability: $r = 0.64$). For each rater's category, we estimated the percentage of participants using a particular strategy. We then averaged the percentage of participants using each strategy across raters. On average, participants made their decisions (1) intuitively ($M = 11.8\%$: rater 1: 9.7%; rater 2: 16%; rater 3: 9.7%), (2) based on the advisers' trustworthiness ($M = 55.9\%$: rater 1: 54.8%; rater 2: 51.6%; rater 3: 61.3%), or (3) based on the advisers' advice ($M = 32.3\%$: rater 1: 35.5%; rater 2: 32.3%; rater 3: 29%). Thus, the majority of our

participants (88.2%) explicitly reported to have made their decisions in the TAG based on the adviser's trustworthy character and advice.

**Neuroimage acquisition**. Data were collected with a Siemens MAGNETOM TRIO 3 Tesla scanner at the Freie Universität Berlin. The fMRI scans consisted of an average of 360 contiguous volumes per run (axial slices, 37; slice thickness, 3 mm; interslice gap, 0.6 mm; repetition time (TR), 2000 ms; echo time (TE), 30 ms; flip angle, 70°; voxel size, $3.0 \times 3.0 \times 3.0$ mm$^3$; field of view (FOV), $192 \times 192$ mm$^2$). High-resolution structural images were acquired through a 3D sagittal T1-weighted MP-RAGE (magnetization prepared-rapid gradient echo) (sagittal slices, 176; TR, 1900 ms; TE, 2.52 ms; slice thickness, 1.0 mm; voxel size, $1.0 \times 1.0 \times 1.0$ mm$^3$; flip angle, 9°; inversion time, 900 ms; FOV, $256 \times 256$ mm$^2$).

**Neuroimage preprocessing**. Neuroimaging data analyses were performed on SPM12 (v. 6905; http://www.fil.ion.ucl.ac.uk/spm/software/spm12/) in MATLAB 2016b (The Mathworks, Natick, MA; http://www.mathworks.com/). The functional images were slice-timing corrected, corrected for voxel displacement using field maps and realigned for head movement correction to the mean image. Using the unified segmentation procedure[62], functional images were co-registered to their structural images and subsequently normalized into MNI (Montreal Neurological Institute) space using deformation fields (resampling voxel size: $2 \times 2 \times 2$ mm$^3$). Finally, functional images used for univariate analyses were spatially smoothed using a Gaussian filter ($8 \times 8 \times 8$ mm$^3$ full-width at half-maximum) to decrease spatial noise. Movement outliers were identified and excluded if head movements/translations were above 3 mm/rad. One run of two participants met these criteria and was therefore excluded from all analyses.

**Behavioral analyses**. Differences in advice-taking behaviors between honest and dishonest advisers were tested with a paired $t$ test (two-sided). The effect size (Cohen's d) was computed as follows: $(M_h - M_d)/\sigma_{diff}$, where $M_h$ and $M_d$ are the average advice-taking behaviors for the honest and dishonest advisers, respectively, and $\sigma_{diff}$ is the standard deviation of the behaviors' differences. A generalized mixed-effects logistic regression was implemented to investigated whether trial-by-trial advice-taking behavior was predicted by the adviser's honesty irrespective of the benefits associated with the act of trust. A model with the following four regressors was built to predict trust in the adviser's advice (1 = trust; 0 = distrust): one regressor coding for the adviser's honesty, one for the advised card, one for the advised number, and one for the feedback in the previous trial played with the current adviser. Random-effects structure was based on a 'maximal' approach with by-subject and by-item random intercepts and slopes[63]. $P$ values were computed with a likelihood-ratio test by comparing the full model with the same model without the fixed effect of interest, but that is otherwise identical in random-effects structure[63]. A mixed-effects regression was further fitted to the difference of advice-taking behaviors toward honest and dishonest advisers with run as fixed-effects time variable and subject as random intercept to test the increase of trust difference over time. Two similar mixed-effects regression models were then separately fitted to each advice-taking behavior toward honest and dishonest advisers in order to examine increases/decreases of trust in the two advisers over time. To test whether trustworthiness relates to subsequent economic trust decisions in a different social context, advice-taking behavior in the TAG was correlated with the amount of money invested in the TG (Spearman's correlations). To further probe that trust decisions in the TG followed from participant's impressions of the partner's trustworthiness in the TAG and were not simply reflecting a repaying behavior, correlation analyses were performed between average winnings in the TAG and money invested in the TG.

**Univariate and ROI analyses**. Two GLMs with eight regressors of interest (two for each task phase) on the first level were defined for both univariate and multivariate analyses of fMRI data to be able to estimate beta parameters that uniquely capture neural signals related to trustworthiness and value encoding, respectively. GLM1 consisted of the following regressors: two regressors for the advisor phase, two regressors for the advice phase, two regressors for the decision phase and two regressors for the feedback phase coding the adviser's trustworthiness (honesty/dishonesty). GLM2 entailed the same regressors as GLM1, with the exception that the two regressors for the feedback phase coded value information (winning/loss).

Control analyses were performed to check that the neural signatures of trustworthiness were not confounded by other factors. In particular, we re-ran GLM1 adding further regressors and parametric modulators to account for variance that might be due to risk and congruency effects. To control for risk, two orthogonal parametric modulators were added to the two regressors coding honesty and dishonesty in the feedback phase; namely, a first-order term for reward probability given the adviser's advice and a second-order term for reward variance (i.e., the mean-squared deviation from expected outcome), which is quadratic in reward probability $p$ and refers to the expected risk given the adviser's advice[64]. Second, to control for contingency effects (i.e., informational deviance between the adviser's advice and the actual card number on the advised card), we added a regressor coding for all feedback phases (i.e., across advisers) with duration 1 s and degrees of congruency (continuous variable) as parametric modulator. Further, to control that the observed neural signatures for honesty and dishonesty

are specific to social information processing and are not confounded by neural signatures of nonsocial information processing, a single parametric GLM was built with only one feedback regressor and two categorical, parametric modulators, that is, first one coding for value information (1 = winning, −1 = loss) and then one coding for the adviser's trustworthiness (1 = honesty, −1 = dishonesty). The two parametric modulators were orthogonalized to be able to capture unique variance related to social information processing.

Finally, to separately investigate brain activations for responses to positive and negative outcomes when interacting with honest and dishonest advisers, and to analyze the modulation of neural responses to value information by honesty and dishonesty, GLM3 was defined encompassing a total of 10 regressors of interest. All task phases had the same regressors as GLM1 and GLM2, except for the feedback phase, for which four regressors were defined coding winnings and losses received when advised by honest and dishonest advisers, separately. In all GLMs, conditions were modeled as events using a stick function (i.e., setting the duration of each condition to 0).

Motion parameters were further included as regressors of no-interest in all GLMs. A temporal high-pass filter with a cutoff of 128 s was applied for all GLMs. Results were whole-brain corrected for multiple comparison using a voxel-level threshold of $p < 0.001$ and a $FWE_c$ corrected threshold of $p < 0.05$[65]. The ROI analysis for the OFC (area s32) to post-hoc examine the modulation of positive outcomes by honesty was based on the probabilistic map provided by the SPM Anatomy toolbox, v. 2.2[66].

**Multivariate voxel pattern analyses**. Decoding analyses to investigate the neural representations of trustworthiness (honesty/dishonesty) and value (winnings/losses) information were performed using a linear SVM algorithm for binary classification and a whole-brain searchlight approach with a searchlight's radius size of 10 mm. Applying an LOROCV, the SVM was trained on all but one run and tested on the left-out run. This procedure was repeated $n$ times with $n = 5$ (total number of runs) and the algorithm's cross-validated accuracy was computed. To decode character information related to the advisers' trustworthiness, β images from the feedback phase of GLM1 (fitted to unsmoothed, normalized brain images) were used. To decode value information related to winnings and losses, β images from the feedback phase of GLM2 (fitted to unsmoothed, normalized brain images) were used. Searchlight decoding analyses were applied to all voxels within the whole-brain gray matter probability mask provided by SPM and thresholded at 0.1.

Decoding generalization of the trustworthiness and value decoding networks was tested with a classification analysis using an LOSOCV approach in which the SVM was trained on $z$-scored average β images of all but one participant and tested on the left-out participant. Cross-validated accuracy of the group-level classification was tested for significance running a permutation test with 10,000 permutations ($n\_perm$). In each permutation, the SVM was trained on randomly permuted labels using the same LOSOCV approach of the true classification model. The sum of models trained on permuted labels that performed better than the true model was then computed ($p\_models$). The nonparametric $p$ value was assessed including the observed statistics according to the following formula[67]: $(1 + p\_models)/(1 + n\_perm)$. Multivariate prediction analyses to predict subsequent, economic trust decisions in the TG (individual averages of money entrusted to the advisers) from the trustworthiness and value decoding networks were based on the same LOSOCV procedure and permutation test but used support vector regression for prediction of continuous variables.

Decoding analyses were run using The Decoding Toolbox TDT, v. 3.99[68] and custom MATLAB scripts.

**Meta-analytic functional decoding**. To characterize the functional specification of the trustworthiness decoding network, a meta-analytic image decoding analysis was performed using the Neurosynth Image Decoder (neurosynth.org)[29]. The Neuro-synth Image Decoder allows to quantitatively estimate the representational similarity between any task-based activation pattern and meta-analytical activation patterns associated with particular terms and generated based on brain images in the Neurosynth database[69]. Similarity was computed as Pearson's correlations across all voxels between the task-based and the meta-analytical maps. We selected meta-analytic maps based on 12 different terms to test the specific a priori hypothesis that the trustworthiness decoding map more likely related to functional roles in the social domain as opposed to the reward, risk, and congruency domains. It has to be noted that the observed correlations are relatively small but in line with previous research[70]. Moreover, while the analysis is quantitative, the conclusions that can be drawn are descriptive in nature, as there is no inference statistics that tested whether any of the observed correlation coefficients is significantly higher than the others.

**Task-dependent functional connectivity analyses**. To test the information flow between the VMPFC underlying honesty signals and any regions across the whole brain, a task-dependent functional connectivity analysis was implemented using a whole-brain psychophysiological interaction (PPI)[71] analysis with seed region (10 mm radius) around the VMPFC peak coordinates yielded by the univariate contrast. The PPI-GLM consisted of a task regressor, a physiological regressor entailing deconvolved blood-oxygen-level-dependent signal from the seed region

and a regressor for the interaction term with movement parameters as regressors of no interest. Significant connectivity was assessed with a voxel-level threshold of $p < 0.001$ and an FWE cluster-level threshold of $p < 0.05$ within the ROI[72].

**Labeling and data visualization**. The SPM Anatomy toolbox v. 2.2[66] and MRIcron (http://people.cas.sc.edu/rorden/mricron/install.html/) were used for anatomical labeling. MRIcroGL (https://www.mccauslandcenter.sc.edu/mricrogl/home/) was used for brain visualizations.

**Reporting summary**. Further information on research design is available in the Nature Research Reporting Summary linked to this article.

## Data availability
The data that support the findings of this study will be provided to all readers upon request.

## Code availability
All relevant MATLAB code is available from the corresponding author upon request.

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

## Acknowledgements

F.M. was supported by the International Max Planck Research School on the Life Course (LIFE), and S.Q.P. by the German Research Foundation Grants INST 392/125-1 (SFB TRR 134 (C07)), PA 2682/1-1, PA 2682/2-1, and by a grant from the German Ministry of Education and Research (BMBF) and the State of Brandenburg (DZD grant FKZ 82DZD00302).

## Author contributions

G.B. and S.Q.P. designed the experiment; G.B. programmed the task; G.B. and F.M. collected the data; G.B. analyzed the data; G.B., F.M., and S.Q.P. wrote the manuscript.

## Competing interests

The authors declare no competing interests.
