## [Transparent Peer Review File · Nature Communications]

Reviewers' comments:

Reviewer #1 (Remarks to the Author):

This is a very interesting paper that presents behavioral and neural results on trustworthiness from an "advice game" (the TAG). In this game, participants choose one of two cards, with the selection of the higher card yielding monetary payouts for participants. Participants had initially no information about the cards, but advisors gave information about one of two cards. Importantly, this information could either be correct (honest), or incorrect (dishonest) and participants could make their decisions based on this information alone. BOLD responses were recorded while subjects played the TAG game and the fMRI analyses focus on the feedback phase that allows an elegant dissociation between trustworthiness and monetary gain. After scanning, participants played a trust game with the advisors. Results indicate that advice was taken from trustworthy advisors, and that trustworthy advisors received a larger transfer in the trust game. MVPA identifies two different networks, one for honesty (dlPFC, IPS, PCC) and one for winning/losing (striatum, ACC). Furthermore, honesty modulated feedback-related activity in OFC.

In my view, this is a very interesting paper that yields some potentially important results. After reading the paper, however, some important aspects of the experimental design and analysis approaches are still unclear to me and I am asking for clarification below.

TAG and Advisors. 1. What is the exact definition of a dishonest adviser (especially in terms of how this was programmed), e.g., was advice incorrect on a percentage of trials, or on all trials? It is also unclear, from the perspective of the participants, what incentive advisers had to be dishonest other than being purely anti-social or spiteful, or being incompetent and making frequent mistakes. How was this explained to subjects in the instructions and what did subjects believe?
2. It is unclear how the 8 advisers differed in their level of honesty? Was the distribution simply 4 advisers were honest, 4 were dishonest, or did they differ in honesty levels in more subtle ways?
3. Do you have a measure from an exit questionnaire about what subjects believed about the advisers in terms of the questions above, as well as whether they were real people, their level of trustworthiness, etc.? I find this important in experiments that use deception.
4. Given your statement on line 100 "... advice was not which was the winning card participants should choose, but rather additional information about the number of one of the two cards ...", what exactly does it mean to "trust the advice of an honest adviser" (line 100). How was trust operationally defined in the context of the TAG, e.g., trust = selecting the card that the adviser gave information about, or by identifying the probability of winning based on the information provided (see my comment on probability below)? What was the DV in the regression model in table 1, in other words, what does trust / distrust (line 454) mean in this respect?

Probability. The game is similar to Preuschoff et al., 2006/2008 in some important respects, such that a probability of winning can be calculated based on knowing one card. For instance, in the example you show in Fig.S1 the adviser indicates that one of the cards is an 8, which has a very high likelihood of being the winning card. This information has been shown by Preuschoff et al. to fundamentally underlie decisions in this task, but seems to be ignored in all analyses here. So, basically, assuming that subjects know one card (even with some probability of this information being incorrect), the optimal strategy should be to compute the probability of this card being the winning one. It seems to me that this is what some subjects did – do you have any information on this in your data? I also think that this is the optimal choice strategy, contrary to your claim on line 423 (but maybe there is something about the design that I am missing).

I understand that the focus here is on the probability that the advice is correct, however, risk and risk prediction errors are an important next step in the decision process (after having assessed the reliability of the advice). If you ignore this aspect, are you not leaving important factors unexplained? Was this controlled for in the experimental design and explicitly communicated to participants (i.e. did I miss anything), or were risk and risk prediction errors explicitly modeled? If not, they could be computed and added as a parametric modulator to the fMRI models.

Multiple GLMs. 1. Are multiple GLMs necessary? It would be very helpful to provide a more detailed explanation of why 4 GLMs are needed - it seems to me that GLM3 or 4 cover all possible combinations of regressors and could be used to extract all the results reported in the paper. Please explain, why GLM1 and GLM2 are necessary (used for SVMs, but in these simple models you leave out an important variable and are then not controlling for the variance associated with one category that clearly influences decision making). Finally, it is unclear to me how GLM4 is different from GLM3 – including a categorical variable as a parametric modulator should in principle be equivalent to binning variables into categories, as they both model the interaction between honesty and dishonesty.

2. If you decide to use multiple GLMs, please make sure to indicate which GLM yielded which results throughout the paper, as this is currently unclear from the results section and figure captions.

Minor comments.

Line 425, consider “except” instead of expect.

Line 141 – There seems to be a relationship between adviser honesty and winnings in the TAG ($p = 0.072$). How does this relationship look like if you correlate only the winnings from each of the conditions (winnings in TAG during honest advisor correlated with trust in TG / TAG winnings during dishonest advisor correlated with trust in TG) and not the average winnings? Moreover, is there a significant difference in participants’ winnings for honest/dishonest advisors in the TAG? This may be purely semantics, but I kept getting confused between trust in the TAG and trust in the TG. Would it make sense to refer to behavior in the TAG as advice-taking throughout the paper - as you do in figure 2, while in the text you call it trust? Also, it was unclear to me how “trustworthiness perceptions” (line 118,) and “strategy” (line 123) were measured (in an exit questionnaire?).

Reviewer #2 (Remarks to the Author):

The goal of this study seems to be to investigate the role of honesty of an advisor for subsequent trust towards that advisor. Participants are trained to guess the higher of two cards. An honest or a dishonest advisor provide information about one of the two cards. The subjects more often take advice from the honest than the dishonest advisor and advice taking correlates with subsequent trust in both advisors. Only the advice taking task was scanned. Multivariate analyses decode honest from dishonest advice in a network of posterior cingulate cortex and frontoparietal regions and wins from losses in medial prefrontal cortex and the striatum (among other regions). Moreover, univariate analyses suggest that dorsolateral prefrontal and parietal regions are more active for dishonest than honest advice whereas VMPFC is more active for honest than dishonest advice and connectivity of the VMPFC with the TPJ correlates with trust in the trust game.

The question of how social information and experiences affect subsequent trust is important and timely. Accordingly, this is not the first paper that addresses it at the neural level (e.g., Delgado et al., 2005, Nat Neurosci; Fareri et al., 2012, Front Neurosci and 2015, JNeurosci; FeldmannHall et al., 2018, PNAS; King-Casas et al., 2005, Science; Maurer et al., 2018, Cognition; Phan et al., 2010, PNAS; van 't Wout & Sanfey, 2008, Cognition; Wardle et al., 2013, PLOS One; none of these are cited). Moreover, the authors do not fully address the question. They take honesty as a proxy for trustworthiness at the behavioral level and primarily focus on the distinction between honest and dishonest feedback at the neural level. The neural insights are limited by the fact that the authors did not scan the trust game.

Issues

The authors assume that honesty increases trustworthiness, which in turn increases trust. In several instances they even equate honesty with trustworthiness. While this assumption may seem reasonable, honesty and trustworthiness are not the same thing and we do not know to what

degree they are related here. Moreover, some of the literature cited above studies the link between trustworthiness and trust more directly. Together with the lack of scanning of the trust game (which would have allowed to test whether trust vs. no-trust decisions can be decoded from honest vs. dishonest advice) one is left with the impression that the current framing is an afterthought and that there could have been an experimental design that investigates the question much more directly, both at the neural and the behavioral level.

Relatedly, the contrast of honest vs. dishonest advice may not only concern perception of (dis)honesty but involve potential confounds such as congruence vs. incongruence of informational inputs or feedback with more or less need for mentalizing. Another potential confound is risk – following the advice of honest advisors is less risky than following the advice of dishonest advisors (which is relevant because in real life low risk advisors would be associated with higher utility in typically risk averse individuals). This, together with the observation of a trend-level correlation between trust and value in the advice game and the activation of VMPFC by honesty (which simply could reflect the possibility that subjects found honesty more valuable than dishonesty) indicates that the separation between value and trust may have been far less complete than the authors suggest. Together, these issues constitute a major limitation to the conclusions that can be drawn from this study.

Whether subjects follow advice or not may reflect susceptibility to social influence rather anything in particular about honesty. What was the correlation in propensity to follow advice for honest and dishonest advisors and does the strength of this relation predict trust?

More information about the (generalized) mixed-effects models is needed: Was participant-level intercept the only random effect included? What approach to obtain p values was used? For mixed-effects models, much hinges on the specification of the random effects and thus it is absolutely crucial to know the details (e.g., whether the models were "maximal" in the sense of Barr et al., 2013).

The rationale for, and selection of, the OFC ROI is not sufficiently justified. There is no a priori hypothesis for OFC (except if honesty were valuable but this would require different framing and one may just as well include the striatum into the ROI).

The decoding analyses are not described in sufficient detail. For example, what algorithm was used, what was the duration of the analyzed time window what was the radius of the searchlight? Moreover, presumably the visual similarity between winning and losing trials was higher than that of honest and dishonest feedback trials, which could explain the higher decoding accuracy for the former than the latter. This should be described and acknowledged.

Figures 4 should provide a separate bar plot for honest and dishonest feedback, figure 5 should show overlap of A and B (and possibly unify labels).

The study used deception in multiple ways, which is currently not acknowledged sufficiently. On the one hand it is unclear whether participants truly believed that others were present and actually provided advice and it is unclear whether participants indeed did not perceive reputation concerns. On the other hand, and more importantly, disentangling value from honesty is likely to have resulted in cards not being drawn according to a uniform distribution, particularly in the honest condition. This could easily have led to additional cognitive processes other than the perception of honesty vs. dishonesty. Realized probabilities in the two conditions should be specified and the possibility for confound discussed.

Response to Reviewers

Reviewer #1 (Remarks to the Author):

This is a very interesting paper that presents behavioral and neural results on trustworthiness from an “advice game” (the TAG). In this game, participants choose one of two cards, with the selection of the higher card yielding monetary payouts for participants. Participants had initially no information about the cards, but advisers gave information about one of two cards. Importantly, this information could either be correct (honest), or incorrect (dishonest) and participants could make their decisions based on this information alone. BOLD responses were recorded while subjects played the TAG game and the fMRI analyses focus on the feedback phase that allows an elegant dissociation between trustworthiness and monetary gain. After scanning, participants played a trust game with the advisers. Results indicate that advice was taken from trustworthy advisers, and that trustworthy advisers received a larger transfer in the trust game. MVPA identifies two different networks, one for honesty (dlPFC, IPS, PCC) and one for winning/losing (striatum, ACC). Furthermore, honesty modulated feedback-related activity in OFC.

In my view, this is a very interesting paper that yields some potentially important results. After reading the paper, however, some important aspects of the experimental design and analysis approaches are still unclear to me and I am asking for clarification below.

We thank the reviewer for this appreciation and constructive suggestions of our work.

TAG and Advisers. 1. What is the exact definition of a dishonest adviser (especially in terms of how this was programmed), e.g., was advice incorrect on a percentage of trials, or on all trials? It is also unclear, from the perspective of the participants, what incentive advisers had to be dishonest other than being purely anti-social or spiteful, or being incompetent and making frequent mistakes. How was this explained to subjects in the instructions and what did subjects believe?

We thank the reviewer for the opportunity to clarify this point. Dishonest advisers sent inaccurate information on all trials (i.e., 100%), whereas honest advisers sent accurate information on all trials. There was no explicit description about advisers' behavior, instead, participants could learn it during the social interaction. The take advice game (TAG) served as a reputation-formation phase, in which participants could learn the adviser's honesty. Based on how the adviser behaved in the TAG, participants made their subsequent decisions in the trust game (TG). This was consistent with the idea that honesty functions as an antecedent of trustworthiness and trusting behavior (Ashton, Lee, & de Vries, 2014; Mayer, Davis, & Schoorman, 1995; Rousseau, Sitkin, Burt, & Camerer, 1998). Participants were told that they were going to play two games with the advisers and that the advisers were informed about that as well. Hence, everybody knew about the social dynamics and the future interactions with the others. Given that the relationship of interdependency between participants and advisers changed across games so that participants depended on the advisers for their monetary payoffs in the TAG, while the advisers depended on the participants for their monetary payoffs in the TG, from the perspective of the participants, advisers in the TAG were motivated to behave honestly in the hope that participants would repay them later on in the TG. However, as there were no direct incentives for being honest during the TAG, there was room left in the decisions

about how to behave in the two games. We now clarified this point in the methods section (lines 449-458, page 25):

“Participants also knew that advisers could help them but did not have any benefits in doing so. However, both partners knew that after the TAG they were going to play a second game (i.e., the TG, see below), in which participants could repay the advisers for their honesty in advice giving. Thus, in the TAG, advisers were motivated to form a good reputation in the hope that participants would repay them later on. To note, however, participants did not promise or commit to repay the advisers for their advice. The dynamics set into motion by this design resembles real-life interactions in which honest behavior (e.g., giving good advice) has often no proximal benefits to an individual but may help her form a good reputation that might turn out advantageous in the future (a possible, distal benefit).”

2. It is unclear how the 8 advisers differed in their level of honesty? Was the distribution simply 4 advisers were honest, 4 were dishonest, or did they differ in honesty levels in more subtle ways?

This is correct, 4 advisers were honest and 4 were dishonest, with 100% contingency. This information is provided in the methods (lines 479-482, page 26):

“we manipulated the advisers’ honesty with the four honest advisers sending accurate information and the four dishonest advisers sending inaccurate information (with 100% contingency)”

3. Do you have a measure from an exit questionnaire about what subjects believed about the advisers in terms of the questions above, as well as whether they were real people, their level of trustworthiness, etc.? I find this important in experiments that use deception.

We thank the reviewer for bringing up this point. We asked participants in an exit questionnaire whether they used any strategy for their decisions during the game and to describe their strategy. Thereby, we were able to capture the motives that guide participants’ behavior in the TAG. Three researchers, who were blind to the study, grouped participants’ free answers into categories. The majority of the participants (88.2%) explicitly reported that they made their decisions in the TAG based on the trustworthiness of the advisers and on their advice. We now report these descriptive analyses in the paper (lines 122-124, page 7):

“Indeed, the majority of our participants (M = 88.2%) explicitly reported in an exit questionnaire (s. Methods) that their decisions were based on the trustworthiness and advice of the advisers.”

We now describe these results in detail in the methods (lines 508-522, pages 27-28):

“Moreover, they were also asked to describe the criteria for their decisions in the TAG (answering the question: “which strategy did you use for your choices in the first game?”). Three researchers blind to the study design and purposes categorized participants’ free answers. The first rater identified three main strategies. The second and third raters identified further subcategories for a total of seven and eight categories, respectively. These could be grouped into the three main strategies of the first rater (averaged inter-rater

reliability: $r = .64$). For each rater's category, we estimated the percentage of participants using a particular strategy. We then averaged the percentage of participants using each strategy across raters. On average, participants made their decisions 1) intuitively ($M = 11.8\%$: rater 1: 9.7%; rater 2: 16%; rater 3: 9.7%), 2) based on the advisers' trustworthiness ($M = 55.9\%$: rater 1: 54.8%; rater 2: 51.6%; rater 3: 61.3%), or 3) on the advisers' advice ($M = 32.3\%$: rater 1: 35.5%; rater 2: 32.3%; rater 3: 29%). Thus, the majority of our participants (88.2%) explicitly reported to have made their decisions in the TAG based on the adviser's trustworthiness character and advice."

4. Given your statement on lines 100 " ... advice was not which was the winning card participants should choose, but rather additional information about the number of one of the two cards ...", what exactly does it mean to "trust the advice of an honest advisor" (lines 100). How was trust operationally defined in the context of the TAG, e.g., trust = selecting the card that the advisor gave information about, or by identifying the probability of winning based on the information provided (see my comment on probability below)? What was the DV in the regression model in table 1, in other words, what does trust / distrust (lines 454) mean in this respect?

Probability. The game is similar to Preuschoff et al., 2006/2008 in some important respects, such that a probability of winning can be calculated based on knowing one card. For instance, in the example you show in Fig.S1 the advisor indicates that one of the cards is an 8, which has a very high likelihood of being the winning card. This information has been shown by Preuschoff et al. to fundamentally underlie decisions in this task, but seems to be ignored in all analyses here. So, basically, assuming that subjects know one card (even with some probability of this information being incorrect), the optimal strategy should be to compute the probability of this card being the winning one. It seems to me that this is what some subjects did – do you have any information on this in your data? I also think that this is the optimal choice strategy, contrary to your claim on lines 423 (but maybe there is something about the design that I am missing). I understand that the focus here is on the probability that the advice is correct, however, risk and risk prediction errors are an important next step in the decision process (after having assessed the reliability of the advice). If you ignore this aspect, are you not leaving important factors unexplained? Was this controlled for in the experimental design and explicitly communicated to participants (i.e. did I miss anything), or were risk and risk prediction errors explicitly modeled? If not, they could be computed and added as a parametric modulator to the fMRI models.

We thank the reviewer for the opportunity to clarify these points. We also assumed that participants would behave according to the optimal strategy described by the reviewer and operationalized advice-taking behavior according to this strategy. As suggested by the reviewer, we now show data that this was indeed the case. To this end, we tested whether the probability of choosing a card is based on the advice received. Because, if participants were computing the probability of a card being the winning one, they should be more likely to pick a card when the adviser had told them that a high number was on that card. Importantly, given the difference in the advisers' honest behaviors, participants' card-choice behaviors should also be sensitive to the adviser's honesty. As the reviewer can see in Fig. S5 participants followed the optimal strategy for both advisers. Moreover, a repeated-measures ANOVA with card number as repeated measure revealed an interaction effect between card number and adviser ($F_{(7,210)}=4.86, p<.0001$). In particular,

participants enacted the optimal strategy more strongly when advised by honest than dishonest advisers, likely because participants did not trust the latter. We now added these results to the revised Methods section (lines 475-492, page 26).

“Advice-taking behavior in the TAG was operationalized as the probability of choosing a card given the informativeness of the advice received. The optimal strategy in the game would be to choose more frequently a card when the adviser communicated that a number bigger than five is on that card but choose the other card when the adviser communicated that a number smaller than five is on that card. Moreover, as we manipulated the advisers’ honesty with the four honest advisers sending accurate information and the four dishonest advisers sending inaccurate information (with 100% contingency), we hypothesized that participants would employ the optimal card-choice strategy differently across honest and dishonest advisers. Analyses of card choice probabilities confirmed our hypotheses (Fig. S5). A repeated-measures ANOVA with card numbers as repeated measure yielded a significant main effect of card number ($F_{(7,210)} = 83.13$; $p < .0001$; $\eta_p^2 = 0.74$) with participants being more likely to choose a card when a number higher than five was said to be on the card and less likely to do so otherwise. Importantly, an interaction effect between card number and advisers was also found ($F_{(7,210)} = 4.86$; $p < .0001$; $\eta_p^2 = 0.14$), suggesting that participants were more likely to follow such a strategy when advised by honest advisers, likely because participants learnt that they could trust the informativeness of the advice of honest advisers.”

Fig. S5:

Fig. S5. Card-choice probability analysis.

Advice-taking behavior in the take advice game (TAG) was operationalized as the probability of choosing a card given the informativeness of the advice received. The optimal strategy in the game would be to choose more frequently a card when the adviser communicated that a number bigger than five is on that card but choose the other card when the adviser communicated that a number smaller than five is on that card. Moreover, as we manipulated the advisers’ honesty, participants should have employed the optimal card-choice strategy differently for honest and dishonest advisers. In particular, they should have used this strategy more loosely for dishonest advisers compared to honest advisers. Analyses of card choice probabilities

confirmed our hypotheses. Depicted is the card-choice probability for each advised card number across participants.

We also thank the reviewer for mentioning the risk dimension as possible confounding variable in our design. First, we would like to stress that previous work has already shown that trust differs from risk and propensity to trust cannot be explained by risk preferences (Aimone & Houser, 2011; Ashraf, Bohnet, & Piankov, 2006; I. Bohnet, Greig, Herrmann, & Zeckhauser, 2008; Iris Bohnet & Zeckhauser, 2004). However, we agree with the reviewer that given our task, some neural activity related to risk prediction errors during the feedback phase might have confounded our results. To address this concern about whether the observed neural patterns of trustworthiness are indeed specific to perceptions of others' social behavior, we employed two approaches.

First, we re-ran our GLM analysis adding parametric modulators to control for congruency and risk effects. These analyses revealed that our results hold also after controlling for those possible confounds (Fig. S2).

Second, we performed a meta-analytic functional decoding analysis to quantitatively evaluate the representational similarity between the neural patterns of our trustworthiness decoding network and meta-analytic neural patterns associated with specific terms in the fMRI literature. This method has been increasingly used in the neuroimaging literature to characterize the specific functional roles and associations of two or more neural maps (Ashar, Andrews-Hanna, Dimidjian, & Wager, 2017; Gao et al., 2018). This way, it is possible to characterize the functional role of a study's neural patterns by a quantitative comparison with previously observed neural patterns associated with certain cognitive functions. We thus selected 12 different terms, among which 'risk', 'risk taking', 'congruent', and 'incongruent' (s. Fig. S3), and estimated the similarity of these meta-analytic maps with our results. Results show that our trustworthiness decoding network was associated neither with risk nor with congruency. Instead, it revealed a strong similarity with neural patterns associated with mentalizing, judgments, social cognition and social interactions. We report the results of these analyses in the results and methods sections.

In Results (lines 187-205, pages 11):

"Finally, we set out to characterize the peculiar functional associations of the trustworthiness decoding network. We first ran GLM analyses to control for possible confounds of the observed neural patterns. In particular, we computed another GLM1 adding parametric modulators to the feedback phase for risk (as mean squared deviation from the expected outcome given the adviser's advice) and congruency (as deviance of the adviser's advice from the actual card number on the advised card). These analyses revealed that our results hold also after controlling for these factors (Fig. S2). Second, using meta-analytic functional decoding (neurosynth.org) (Yarkoni, Poldrack, Nichols, Van Essen, & Wager, 2011), we quantitatively evaluated the representational similarity of the trustworthiness decoding network with neural activation patterns associated with specific psychological components. In particular, we compared the neural signatures of trustworthiness in our study against reverse inference meta-analytic neural patterns of neural images of previous studies stored in the Neurosynth database and associated with particular psychological terms. For this

analysis, we chose twelve terms associated with the social and nonsocial domains, such as social cognition, theory of mind, rewards, congruency and risk (Fig. S3). Results demonstrate that the trustworthiness decoding network was preferentially associated with psychological terms related to mentalizing and social cognition (Fig. S3), validating the ability of our task in singling out neural patterns that likely underlie the formation of trustworthiness beliefs about the advisers.”

Fig. S2:

Fig. S2. Control GLM analysis.

Comparison of GLM1 coding for honesty and dishonesty (red), and the control GLM that further controlled for risk and congruency effects (green). The two GLMs yielded similar results. In yellow are the overlaps depicted.

GLM, general linear model; IPS, intraparietal sulcus; DLPFC, dorsolateral prefrontal cortex; PCC, posterior cingulate cortex

Fig. S3:

Fig. S3. Meta-analytic functional decoding analysis.

To test the functional specificity of the trustworthiness decoding network, we performed a meta-analytic functional decoding analysis. Using the Neurosynth database, we evaluated the representational similarity between the neural patterns of our trustworthiness map and meta-analytic neural patterns associated with specific terms in the fMRI literature. This way, it was possible to characterize the functional role of our neural patterns by a quantitative comparison with previously observed neural patterns associated with certain cognitive functions. We selected twelve different terms in the social, value, risk and congruency domain. Results show that the neural signatures of the trustworthiness decoding network reveal stronger similarity with neural patterns associated with mentalizing, judgments, social cognition and social interactions than with other cognitive functions. Values on the spider plot (-0.03 — 0.17) represent Pearson's correlation coefficients.

In Methods (lines 577-589, page 30):

“Control analyses were performed to check that the neural signatures of trustworthiness were not confounded by other factors. In particular, we re-ran GLM1 adding further regressors and parametric modulators to account for variance that might be due to risk and congruency effects. To control for risk, two orthogonal parametric modulators were added to the two regressors coding honesty and dishonesty in the feedback phase; namely, a 1st order term for reward probability given the adviser's advice and a 2nd order term for reward variance (i.e., the mean squared deviation from expected outcome), which is quadratic in reward probability p and refers to the expected risk given the adviser's advice (Preuschoff, Bossaerts, & Quartz, 2006). Second, to control for contingency effects (i.e., informational deviance between the adviser's advice and the actual card number on the advised card), we added a regressor coding for all feedback phases (i.e., across advisers) with duration 1s and degrees of congruency (continuous variable) as parametric modulator.”

In Methods (lines 633-640, page 32):

“Meta-analytic functional decoding. To characterize the functional specification of the trustworthiness decoding network, a meta-analytic image decoding analysis was performed

using the Neurosynth Image Decoder (neurosynth.org; Yarkoni et al., 2011). The Neurosynth Image Decoder allows to quantitatively estimate the representational similarity between any task-based activation pattern and meta-analytic activation patterns associated with particular terms and generated based on brain images in the Neurosynth database (Kriegeskorte, Mur, & Bandettini, 2008). Similarity was computed as Pearson's correlations across all voxels between the task-based and the meta-analytical maps."

Multiple GLMs. 1. Are multiple GLMs necessary? It would be very helpful to provide a more detailed explanation of why 4 GLMs are needed - it seems to me that GLM3 or 4 cover all possible combinations of regressors and could be used to extract all the results reported in the paper. Please explain, why GLM1 and GLM2 are necessary (used for SVMs, but in these simple models you leave out an important variable and are then not controlling for the variance associated with one category that clearly influences decision making). Finally, it is unclear to me how GLM4 is different from GLM3 – including a categorical variable as a parametric modulator should in principle be equivalent to binning variables into categories, as they both model the interaction between honesty and dishonesty.

We thank the reviewer for raising these concerns. Our design elegantly allowed us to disentangle trustworthiness from value information. However, to investigate their neural signatures, it was necessary to build two GLMs, as the same feedback phase had to be coded differently. It was not possible to have in the same GLM both regressors coding the adviser's honesty across value information, and value information across adviser's honesty because this would have made the betas impossible to estimate. Finally, to examine the interaction between these two types of information, namely how positive and negative outcomes were processed when interacting with honest and dishonest advisers, GLM3 was built. Since we showed on the behavioral level that trustworthiness and value information could be successfully disentangled, and that value information did not influence participants' decisions ($\beta = -0.001$; $SE = 0.07$; $95\% CI = [-0.14, 0.14]$; $p = .980$; Tab. 1), the definition of two GLMs that separately coded the adviser's trustworthiness and value information is reasonable. Moreover, not only our MVPA results indicate that the SVM was able to capture the specific neural dynamics of trustworthiness and value information processing, we also showed that the trustworthiness decoding network, but not the value decoding one, could successfully predict future trust, indicating some degree of specificity of these neural patterns in representing trustworthiness information about others that guides future behavior. Following the reviewer's suggestion, we now provide an explanation of the use of these GLMs in the Methods (lines 569-562, pages 29-30):

"Two general linear models (GLMs) with eight regressors of interest (two for each task phase) on the first level were defined for both univariate and multivariate analyses of fMRI data to be able to estimate beta parameters that uniquely capture neural signals related to trustworthiness and value encoding, respectively"

Finally, we completely agree with the reviewer that GLM3 and GLM4 are exactly the same and now collapsed the two GLMs into one as follows (lines 590-593, page 30):

"Finally, to separately investigate brain activations for responses to positive and negative feedback when interacting with honest and dishonest advisers, and to analyze the honesty modulation of brain regions processing feedback information, GLM3 was defined"

2. If you decide to use multiple GLMs, please make sure to indicate which GLM yielded which results throughout the paper, as this is currently unclear from the results section and figure captions.

We now indicate which GLM yielded which results throughout the paper and in figure captions.

Minor comments.

Lines 425, consider “except” instead of expect.

We thank the reviewer for spotting this typo. We corrected it accordingly.

Lines 141 – There seems to be a relationship between adviser honesty and winnings in the TAG ($p = 0.072$). How does this relationship look like if you correlate only the winnings from each of the conditions (winnings in TAG during honest advisor correlated with trust in TG / TAG winnings during dishonest advisor correlated with trust in TG) and not the average winnings? Moreover, is there a significant difference in participants’ winnings for honest/dishonest advisors in the TAG?

We thank the reviewer and added the suggested analyses. The trend seemed indeed driven by two extreme values ($> 2 SD$). Nonparametric analyses (Spearman correlations) revealed no significant relationships. Also, no significant correlations were found when correlating winnings from each of the conditions separately.

Lines 139-142, page 8:

“advice-taking behavior in the TAG correlated with subsequent, economic trust decisions in the TG on average ($\rho_{(29)} = .39$; $p = .031$), and separately for both honest ($\rho_{(29)} = .41$ $p = .021$) and dishonest advisers ($\rho_{(29)} = -.37$; $p = .040$).”

And in lines 144-147, page 8:

“the amount of money shared with the advisers in the TG did not significantly correlate with participants’ monetary winnings in the TAG either on average ($\rho_{(29)} = .17$; $p = .350$) or separately for the two advisers (honest adviser: $\rho_{(29)} = .30$; $p = .106$; dishonest adviser: $\rho_{(29)} = .01$; $p = .978$)”

Finally, we did not find an overall significant effect of winnings, although participants won more when playing with honest advisers.

Lines 151-155, page 9:

“Finally, we checked the proportion of feedback received by our participants. Participants received on average the same amount of positive and negative feedback (mean difference = $0.0013 \pm SD = 0.07$; $t_{(30)} = 0.11$; $p = 0.916$), despite more positive feedback for honest than dishonest advisers (honest advisers: $M = 63.5\% \pm SD = 7.4$; dishonest advisers: $M = 56.7\% \pm SD = 5.0$; $t_{(30)} = 4.09$; $p < 0.001$).”

This may be purely semantics, but I kept getting confused between trust in the TAG and trust in the TG. Would it make sense to refer to behavior in the TAG as advice-taking throughout the paper - as you do in figure 2, while in the text you call it trust? Also, it was unclear to me

how “trustworthiness perceptions” (lines 118,) and “strategy” (lines 123) were measured (in an exit questionnaire?).

We thank the reviewer for pointing to this and we now unified the terminological use throughout the manuscript. Yes, those were measured in an exit questionnaire as reported in the Methods (lines 502-505, page 27).

Reviewer #2 (Remarks to the Author):

The goal of this study seems to be to investigate the role of honesty of an advisor for subsequent trust towards that advisor. Participants are trained to guess the higher of two cards. An honest or a dishonest advisor provide information about one of the two cards. The subjects more often take advice from the honest than the dishonest advisor and advice taking correlates with subsequent trust in both advisors. Only the advice taking task was scanned. Multivariate analyses decode honest from dishonest advice in a network of posterior cingulate cortex and frontoparietal regions and wins from losses in medial prefrontal cortex and the striatum (among other regions). Moreover, univariate analyses suggest that dorsolateral prefrontal and parietal regions are more active for dishonest than honest advice whereas VMPFC is more active for honest than dishonest advice and connectivity of the VMPFC with the TPJ correlates with trust in the trust game.

The question of how social information and experiences affect subsequent trust is important and timely. Accordingly, this is not the first paper that addresses it at the neural level (e.g., Delgado et al., 2005, Nat Neurosci; Fareri et al., 2012, Front Neurosci and 2015, JNeurosci; FeldmannHall et al., 2018, PNAS; King-Casas et al., 2005, Science; Maurer et al., 2018, Cognition; Phan et al., 2010, PNAS; van 't Wout & Sanfey, 2008, Cognition; Wardle et al., 2013, PLOS One; none of these are cited).

We thank the reviewer for the appreciation of our work and the literature suggestions. Due to limitations in the number of citable papers imposed by the journal (i.e., 70), we were not able to cover all studies investigating trusting behavior with fMRI. We now incorporate some of the suggested studies and cited recent meta-analyses on trust that cover the mentioned fMRI literature.

Moreover, the authors do not fully address the question. They take honesty as a proxy for trustworthiness at the behavioral level and primarily focus on the distinction between honest and dishonest feedback at the neural level. The neural insights are limited by the fact that the authors did not scan the trust game.

We thank the reviewer for this point and apologize that our description might have been misleading. The aim of our study is to investigate the neural mechanisms underlying how trust develops from the honesty of an advisor and not the trust decision per se. Thus, we showed that neural signatures of others' honesty and dishonesty (TAG) predicted future behavioral trust (TG) and that stronger integration of honesty signal correlated with higher future trust. The trust game was not in focus but was applied as a read-out for the transfer effect of established trust based on the honest advice.

In the past literature, the suitability of a single-shot TG to investigate impression-based trust has been repeatedly demonstrated (Bonnefon, Hopfensitz, & De Neys, 2013; Burnham, McCabe, & Smith, 2000; De Neys, Hopfensitz, & Bonnefon, 2017). We therefore decided that the single-shot trust game would be the best behavioral measure for our purposes. Unfortunately, a single-shot trust game with a single decision per adviser is not suitable for fMRI analyses due to lack of power (too few trials). But as mentioned before, studying the trust game itself was not the main objective of our study.

However, we value the reviewer's comment that it remains an open question whether these neural signatures of others' honesty and dishonesty are also able to predict neural signatures underlying trust decisions. We now address this as suggestion for future studies in the Discussion (lines 403-407, page 22).

“Although we here showed that trustworthiness-related neural signal successfully predicts individual trust decisions in a future social interaction, future studies are still needed to investigate in a brain-to-brain predictive framework whether these neural signatures of trustworthiness are also able to predict the neural patterns recruited by individual trust decisions.”

The authors assume that honesty increases trustworthiness, which in turn increases trust. In several instances they even equate honesty with trustworthiness. While this assumption may seem reasonable, honesty and trustworthiness are not the same thing and we do not know to what degree they are related here. Moreover, some of the literature cited above studies the link between trustworthiness and trust more directly. Together with the lack of scanning of the trust game (which would have allowed to test whether trust vs. no-trust decisions can be decoded from honest vs. dishonest advice) one is left with the impression that the current framing is an afterthought and that there could have been an experimental design that investigates the question much more directly, both at the neural and the behavioral level.

We thank the reviewer for this comment. Our study investigates the neural mechanism of how honest behavior of another person (TAG) leads to trust that person (TG) in a future, independent context. Previous theoretical work, especially in psychology, has already posited that honesty, among a set of other character traits, is an antecedent of trustworthiness and trusting behavior (Ashton & Lee, 2007; Ashton et al., 2014; Mayer et al., 1995; Thielmann & Hilbig, 2015). However, as the reviewer has correctly pointed out, there has been a lack of empirical evidence in the extant literature about how honesty and trustworthiness are related to each other. Our study is the very first attempt to show this relationship. Moreover, as the central feature of the neural representation of a character trait, such as trustworthiness, is its ability to inform decisions across contexts, we let participants play the TG after the scanning session to specifically test whether neural signatures of honesty-based trustworthiness are able to predict future trust in a new context. We now make this point clearer in the introduction (lines 60-62, page 4):

“In this study, we investigated for the first time whether information about the other’s honest character evokes trustworthiness perceptions that predict future trust in the other.”

Relatedly, the contrast of honest vs. dishonest advice may not only concern perception of (dis)honesty but involve potential confounds such as congruence vs. incongruence of informational inputs or feedback with more or less need for mentalizing. Another potential confound is risk – following the advice of honest advisors is less risky than following the advice of dishonest advisors (which is relevant because in real life low risk advisors would be associated with higher utility in typically risk averse individuals). This, together with the observation of a trend-level correlation between trust and value in the advice game and the activation of VMPFC by honesty (which simply could reflect the possibility that subjects found honesty more valuable than dishonesty) indicates that the separation between value and trust may have been far less complete than the authors suggest. Together, these issues constitute a major limitation to the conclusions that can be drawn from this study.

We agree with the reviewer that our main contrast of interest might be confounded by other factors, such as congruency and risk. As pointed out in the response to the other's reviewer, to address the concern about whether the observed neural patterns of trustworthiness are indeed specific to perceptions of others' social behavior (honesty and dishonesty), we employed two approaches: 1) parametric modulation analyses to control for congruency and risk effects; and 2) a meta-analytic functional decoding analysis to estimate the similarity of our neural patterns with meta-analytic neural maps associated with congruency, risk and value, on one hand, and social cognition, on the other. These analyses revealed that 1) our results hold also after controlling for those possible confounds and that 2) the trustworthiness decoding network identified by our analyses was more strongly related to neural patterns associated with theory of mind, social interactions, judgments and social cognition than with value, risk or congruency (Fig. S2). We now report these results in the manuscript.

In Results (lines 187-205, pages 11):

"Finally, we set out to characterize the peculiar functional associations of the trustworthiness decoding network. We first ran GLM analyses to control for possible confounds of the observed neural patterns. In particular, we computed another GLM1 adding parametric modulators to the feedback phase for risk (as mean squared deviation from the expected outcome given the adviser's advice) and congruency (as deviance of the adviser's advice from the actual card number on the advised card). These analyses revealed that our results hold also after controlling for these factors (Fig. S2). Second, using meta-analytic functional decoding (neurosynth.org)(Yarkoni et al., 2011), we quantitatively evaluated the representational similarity of the trustworthiness decoding network with neural activation patterns associated with specific psychological components. In particular, we compared the neural signatures of trustworthiness in our study against reverse inference meta-analytic neural patterns of neural images of previous studies stored in the Neurosynth database and associated with particular psychological terms. For this analysis, we chose twelve terms associated with the social and nonsocial domains, such as social cognition, theory of mind, rewards, congruency and risk (Fig. S3). Results demonstrate that the trustworthiness decoding network was preferentially associated with psychological terms related to mentalizing and social cognition (Fig. S3), validating the ability of our task in singling out neural patterns that likely underlie the formation of trustworthiness beliefs about the advisers."

In Methods (lines 577-589, page 30):

"Control analyses were performed to check that the neural signatures of trustworthiness were not confounded by other factors. In particular, we re-ran GLM1 adding further regressors and parametric modulators to account for variance that might be due to risk and congruency effects. To control for risk, two orthogonal parametric modulators were added to the two regressors coding honesty and dishonesty in the feedback phase; namely, a 1st order term for reward probability given the adviser's advice and a 2nd order term for reward variance (i.e., the mean squared deviation from expected outcome), which is quadratic in reward probability p and refers to the expected risk given the adviser's advice (Preuschoff et al., 2006). Second, to control for contingency effects (i.e., informational deviance between the adviser's advice and the actual card number on the advised card), we added a regressor coding for all feedback phases (i.e., across advisers) with duration 1s and degrees of congruency (continuous variable) as parametric modulator."

In Methods (lines 633-640, page 32):

“Meta-analytic functional decoding. To characterize the functional specification of the trustworthiness decoding network, a meta-analytic image decoding analysis was performed using the Neurosynth Image Decoder (neurosynth.org; Yarkoni et al., 2011). The Neurosynth Image Decoder allows to quantitatively estimate the representational similarity between any task-based activation pattern and meta-analytic activation patterns associated with particular terms and generated based on brain images in the Neurosynth database (Kriegeskorte et al., 2008). Similarity was computed as Pearson’s correlations across all voxels between the task-based and the meta-analytical maps.”

Whether subjects follow advice or not may reflect susceptibility to social influence rather than anything in particular about honesty. What was the correlation in propensity to follow advice for honest and dishonest advisers and does the strength of this relation predict trust?

We thank the reviewer for suggesting this hypothesis. Our data do not seem to be driven by individual differences in social influence due to the following reasons:

- 1. Our results (in lines with the goal of the study) are based on a significant difference in advice-taking behavior toward honest and dishonest advisers within the same participants. In particular, participants were more likely to take advice from honest than dishonest advisers ($\beta = 0.38$; $SE = 0.11$; $95\% CI = [0.16, 0.59]$; $p < .001$).***
- 2. Had the observed behavior been driven by individual differences in susceptibility to social influence, it should have remained stable over time, which is not the case in our data. Even though participants were equally likely to take advice from both honest and dishonest advisers at the beginning of the task (Fig. 2A), participants’ advice-taking behavior toward honest and dishonest advisers differed increasingly over time ($\beta = 0.01$; $SE = 0.006$; $95\% CI = [0.0001, 0.024]$; $p = .048$), suggesting that in the course of the interaction, participants showed increasingly less similar advice-taking patterns across advisers (Fig. 2A). In particular, this difference was due to a significant increase of distrust in dishonest advisers over the course of the game ($\beta = -0.02$; $SE = 0.007$; $95\% CI = [-0.028, -0.002]$; $p = .021$; see also Fig. 2A), suggesting that participants adapted their behavior based on the honest/dishonest behavior of the adviser. Due to this behavioral adaptation, they ended up taking on average more advice from honest than dishonest advisers ($t_{(30)} = 3.68$; $p < .001$; $95\% CI = [0.03, 0.10]$; $Cohen’s d = 0.7$);).***
- 3. Finally, we asked participants to report their strategy during the TAG. Participants’ subjective reports confirmed that participants made their decisions based on the trustworthiness and advice of advisers (lines 122-124, page 7 and lines 516-522, pages 27-28).***

Although this was not within the scope of our study, it might still be the case that some of the variance in the observed behavioral patterns could be explained by individual differences in susceptibility to social influences. We now explicitly mention this in our discussion (lines 408-410, pages 22-23).

“Further, future studies, especially in advice-taking paradigms, might also consider controlling for individual susceptibility to social influence, which might for instance explain an individual’s propensity to take advice from others.”

More information about the (generalized) mixed-effects models is needed: Was participant-level intercept the only random effect included? What approach to obtain p values was used? For mixed-effects models, much hinges on the specification of the random effects and thus it is absolutely crucial to know the details (e.g., whether the models were "maximal" in the sense of Barr et al., 2013).

We thank the reviewer for this suggestion. We re-ran the models with a maximal approach for random-effects structure. P values were computed using a likelihood-ratio test. No changes in the results could be observed. We now include a detailed description of the mixed-effects models in the methods (lines 553-556, page 29).

“Random-effects structure was based on a ‘maximal’ approach with by-subject and by-item random intercepts and slopes (Barr, Levy, Scheepers, & Tily, 2013). P-values were computed with a likelihood-ratio test by comparing the full model with the same model without the fixed effect of interest but that is otherwise identical in random-effects structure (Barr et al., 2013).”

The rationale for, and selection of, the OFC ROI is not sufficiently justified. There is no apriori hypothesis for OFC (except if honesty were valuable but this would require different framing and one may just as well include the striatum into the ROI).

The ROI analysis on the OFC was a post-hoc analysis to descriptively show how neural responses to positive and negative feedback differ across advisers, as it is already mentioned in our hypothesis on the OFC in the Introduction (lines 62-71, page 4). We now add another justification of this analysis in the Results (lines 286-287, page 18):

“Using an independent ROI in the OFC, we more closely examined in a post-hoc ROI analysis this asymmetric honesty modulation of positive feedback processing”

The decoding analyses are not described in sufficient detail. For example, what algorithm was used, what was the duration of the analyzed time window what was the radius of the searchlight?

We thank the reviewer for highlighting these issues. We now added those details to the Methods (lines 604-615, page 31):

“Decoding analyses to investigate the neural representations of trustworthiness (honesty/dishonesty) and value (winnings/losses) information were performed using a linear support vector machine (SVM) algorithm for binary classification and a whole-brain searchlight approach with a searchlight’s radius size of 10mm. Applying a leave-one-run-out cross-validation (LOROCV), the SVM was trained on all but one run and tested on the left-out run. This procedure was repeated n times with n=5 (total number of runs) and the algorithm’s cross-validated accuracy was computed. To decode character information related to the advisers’ trustworthiness, beta images from the feedback phase of GLM1

(fitted to unsmoothed, normalized brain images) were used. To decode feedback information related to winnings and losses, beta images from the feedback phase of GLM2 (fitted to unsmoothed, normalized brain images) were used.”

And in Methods (lines 596-597, page 30):

“In all GLMs, conditions were modeled as events using a stick function (i.e., setting the duration of each condition to 0).”

Moreover, presumably the visual similarity between winning and losing trials was higher than that of honest and dishonest feedback trials, which could explain the higher decoding accuracy for the former than the latter. This should be described and acknowledged.

We are grateful to the reviewer for the opportunity to address this interesting finding. Indeed, winnings and losses show a higher classification accuracy compared to honesty and dishonesty. However, the visual similarity for the two classification analyses was exactly the same, as the SVM was trained on the same trials and phase in both analyses. What differed, was the categorization of trials (whether win vs. loss or honest vs. dishonest), which are otherwise identical. We rather believe that the lower accuracy for honesty and dishonesty may be due to the nature of the classified category (social vs. nonsocial). Previous studies have pointed to similar accuracy patterns. For instance, classification accuracy for social pain is substantially lower than classification accuracy for physical pain based on neural activity (Woo, Chang, Lindquist, & Wager, 2017). We now acknowledge this (lines 182-186, page 10):

“To note, classification accuracy of value information was much better than classification accuracy of social character information. These results concur with previous findings (Woo et al., 2014) and may hinge on the nature of social concepts, which are distributed neural representations that might be difficult to fully capture using an anatomical-based searchlight approach.”

Figures 4 should provide a separate bar plot for honest and dishonest feedback, figure 5 should show overlap of A and B (and possibly unify labels).

Figure 4 shows now bar plots for honest and dishonest feedback. Finally, Fig. 6A & 6B (previously, Fig. 5) depict the right inferior parietal lobule and left intraparietal sulcus, respectively (see also Tab. S5). As the two activations are in different hemispheres, they do not overlap. Anatomical labeling follows the probabilistic cytoarchitectonic maps provided by the SPM Anatomy toolbox and, for Brodmann areas, the Brodmann atlas provided by MRICron (see in Methods, lines 652-655, page 33).

The study used deception in multiple ways, which is currently not acknowledged sufficiently. On the one hand it is unclear whether participants truly believed that others were present and actually provided advice and it is unclear whether participants indeed did not perceive reputation concerns. On the other hand, and more importantly, disentangling value from honesty is likely to have resulted in cards not being drawn according to a uniform distribution, particularly in the honest condition. This could easily have led to additional cognitive processes other than the perception of honesty vs. dishonesty. Realized

probabilities in the two conditions should be specified and the possibility for confound discussed.

We are grateful to the reviewer for the opportunity to clarify this point. As we pointed out in the response to the other reviewer, participants were explicitly asked to describe the strategy and criteria adopted to make their decisions in an exit questionnaire. The majority of our participants (88.2%) explicitly reported that they made their decisions during the TAG based on the advisers' trustworthiness and their advice.

With respect to the uniform distribution: During the designing of the experiment, we were indeed concerned about the realized probabilities of card drawing in the two conditions, which we solved by using a pseudo-random sampling approach to disentangle value from honesty information. However, we also optimized the sampling procedure so that it approximates chance in both conditions, as would be expected in random drawing. This way, we could avoid that additional cognitive processes might have confounded our results. We now include this in the Methods.

Methods (lines 462-467, page 25):

"Thus, cards were drawn from a uniform distribution with pseudo-random sampling without replacement. The pseudo-random sampling procedure was optimized to have a realized probability of card drawing that approximates chance in both conditions, as would be expected in random drawing. A two-sample Kolmogorov-Smirnov test confirmed that the realized distributions of card numbers did not differ between advisers ($K-S$ test = 0.25; $p = .929$)."

References

- Aimone, J. A., & Houser, D. (2011). Beneficial betrayal aversion. *PLoS One*, 6(3), e17725. doi:10.1371/journal.pone.0017725
- Ashar, Y. K., Andrews-Hanna, J. R., Dimidjian, S., & Wager, T. D. (2017). Empathic Care and Distress: Predictive Brain Markers and Dissociable Brain Systems. *Neuron*, 94(6), 1263-1273 e1264. doi:10.1016/j.neuron.2017.05.014
- Ashraf, N., Bohnet, I., & Piankov, N. (2006). Decomposing trust and trustworthiness. *Experimental Economics*, 9(3), 193-208. doi:10.1007/s10683-006-9122-4
- Ashton, M. C., & Lee, K. (2007). Empirical, theoretical, and practical advantages of the HEXACO model of personality structure. *Pers Soc Psychol Rev*, 11(2), 150-166. doi:10.1177/1088868306294907
- Ashton, M. C., Lee, K., & de Vries, R. E. (2014). The HEXACO Honesty-Humility, Agreeableness, and Emotionality factors: a review of research and theory. *Pers Soc Psychol Rev*, 18(2), 139-152. doi:10.1177/1088868314523838
- Barr, D. J., Levy, R., Scheepers, C., & Tily, H. J. (2013). Random effects structure for confirmatory hypothesis testing: Keep it maximal. *J Mem Lang*, 68(3). doi:10.1016/j.jml.2012.11.001
- Bohnet, I., Greig, F., Herrmann, B., & Zeckhauser, R. (2008). Betrayal Aversion: Evidence from Brazil, China, Oman, Switzerland, Turkey, and the United States. *American Economic Review*, 98(1), 294-310.
- Bohnet, I., & Zeckhauser, R. (2004). Trust, risk and betrayal. *Journal of Economic Behavior & Organization*, 55(4), 467-484. doi:10.1016/j.jebo.2003.11.004
- Bonnefon, J. F., Hopfensitz, A., & De Neys, W. (2013). The modular nature of trustworthiness detection. *J Exp Psychol Gen*, 142(1), 143-150. doi:10.1037/a0028930
- Burnham, T., McCabe, K., & Smith, V. L. (2000). Friend-or-foe intentionality priming in an extensive form trust game. *Journal of Economic Behavior & Organization*, 43(1), 57-73. doi:10.1016/s0167-2681(00)00108-6
- De Neys, W., Hopfensitz, A., & Bonnefon, J. F. (2017). Split-Second Trustworthiness Detection From Faces in an Economic Game. *Exp Psychol*, 64(4), 231-239. doi:10.1027/1618-3169/a000367
- Gao, X., Yu, H., Sáez, I., Blue, P. R., Zhu, L., Hsu, M., & Zhou, X. (2018). Distinguishing neural correlates of context-dependent advantageous- and disadvantageous-inequity aversion. *PNAS*, 115(33), E7680-E7689. doi:10.1073/pnas.1802523115
- Kriegeskorte, N., Mur, M., & Bandettini, P. (2008). Representational similarity analysis - connecting the branches of systems neuroscience. *Front Syst Neurosci*, 2, 4. doi:10.3389/neuro.06.004.2008
- Mayer, R. C., Davis, J. H., & Schoorman, F. D. (1995). An Integrative Model of Organizational Trust. *The Academy of Management Review*, 20(3), 709-734.
- Preuschoff, K., Bossaerts, P., & Quartz, S. R. (2006). Neural differentiation of expected reward and risk in human subcortical structures. *Neuron*, 51(3), 381-390. doi:10.1016/j.neuron.2006.06.024
- Rousseau, D. M., Sitkin, S. B., Burt, R. S., & Camerer, C. (1998). Not So Different after All: A Cross-Discipline View of Trust. *Academy of Management Review*, 23(3), 393-404. doi:10.5465/amr.1998.926617
- Thielmann, I., & Hilbig, B. E. (2015). The Traits One Can Trust: Dissecting Reciprocity and Kindness as Determinants of Trustworthy Behavior. *Pers Soc Psychol Bull*, 41(11), 1523-1536. doi:10.1177/0146167215600530

- Woo, C. W., Chang, L. J., Lindquist, M. A., & Wager, T. D. (2017). Building better biomarkers: brain models in translational neuroimaging. *Nat Neurosci*, *20*(3), 365-377. doi:10.1038/nn.4478
- Woo, C. W., Koban, L., Kross, E., Lindquist, M. A., Banich, M. T., Ruzic, L., . . . Wager, T. D. (2014). Separate neural representations for physical pain and social rejection. *Nat Commun*, *5*, 5380. doi:10.1038/ncomms6380
- Yarkoni, T., Poldrack, R. A., Nichols, T. E., Van Essen, D. C., & Wager, T. D. (2011). Large-scale automated synthesis of human functional neuroimaging data. *Nat Methods*, *8*(8), 665-670. doi:10.1038/nmeth.1635

Reviewers' comments:

Reviewer #1 (Remarks to the Author):

I thank the authors for addressing my comments with such care. I think that the paper has improved significantly and many of my questions have been clarified. I have only a few more requests/comments:

1. In the paragraph starting on line 94, I think it would be beneficial to indicate how subjects earn money in the TAG game to make reading this part easier. It might be unclear to readers at this point (without jumping to the methods section) what exactly winning means in the context of this game (picking the larger card), and how much participants were paid for each win.
2. You might consider noting that the correlation coefficients in Figure S3 are relatively small and depend in part on which map you are comparing it to. I bet if you included the map for the default mode, you would find a higher correlation. In any case, please state that you selected these maps to test specific a priori hypotheses, and that the correlations are relatively small. While the analysis is quantitative, the conclusions drawn from this seem qualitative – compared to other maps of interest, this yields the highest correlation coefficient, but there is no statistic reflecting whether this correlation coefficient is significantly higher than the others.
3. A better citation for the neurosynth decoder might be Rubin et al., Plos Comp. Bio., 2017.
4. Your functional connectivity results agree with a recent paper that show stronger TPJ coupling with a more extensive social cognition network with increasing trust (Engelmann et al., Science Advances, 2019). It might be helpful to incorporate these supportive results in the paragraph starting on line 366.
5. Your analysis reported in Figure S5 and page 26 reports an interaction without a follow-up test of the interaction, so we do not know what drives the interaction. I suggest you run post-hoc tests comparing the average choice probability for honest vs dishonest advisors for cards 1-4 (averaged) and cards 6-9 (averaged). The former should yield dishonest > honest, while the latter should yield honest > dishonest to support your argument. In the absence of this test, the interaction could be driven by the difference between specific cards (e.g., 3 and 9), which would not be so interesting.

Reviewer #2 (Remarks to the Author):

While the revision was responsive it also raised further questions. The advisors provided either correct or incorrect advice in every trial. This is not very realistic and could have raised doubt about the setup. Moreover, the dishonest advisor could provide more less inaccurate advice in the take advice game and be either (i) deliberately misleading or (ii) simply riskier (i.e., with regard to the decision provide similar information as the advisor but with less precision and never match the actual card). In the former case, the optimal response to the dishonest advisors probably would have been to go against the advice, which should have led to behavior following a vertically flipped S-curve in Figure S5 rather than a less pronounced standard S-curve whereas in the latter case it may be rational to keep on relying on the dishonest advice, although this should be confirmed formally. This point should be described and clarified in the Methods (in the paragraph starting at line 475).

The link between the two tasks seems still somewhat overblown. Trust towards the dishonest advisors decreases already in the take advice game. It therefore seems to be more of a validation that trust towards the dishonest advisors is relatively low in the trust game (and that advice taking entails trust such that the labels of the MVPA (from the take advice game predicting subsequent trust in the trust game) might just as well have been "trust high" vs "trust low" rather than "honest" vs. "dishonest"). Accordingly, any trust-reducing behavior other than dishonesty should have yielded similar findings. These points should be acknowledged in the Discussion (e.g., in the

paragraph starting at line 394).

Regarding the possibility to combine GLMs 1 and 2, I may be wrong but could there not be one GLM with one feedback regressor which is modulated by two "parametric" modulators of two levels each, one for honesty and one for gain/loss?

Reviewer #1 (Remarks to the Author):

I thank the authors for addressing my comments with such care. I think that the paper has improved significantly and many of my questions have been clarified. I have only a few more requests/comments:

We thank the reviewer for the appreciation of our work and making those improvements possible.

1. In the paragraph starting on line 94, I think it would be beneficial to indicate how subjects earn money in the TAG game to make reading this part easier. It might be unclear to readers at this point (without jumping to the methods section) what exactly winning means in the context of this game (picking the larger card), and how much participants were paid for each win.

We agree with the reviewer and added this information accordingly (lines 104-105, page 6):

“Participants could win/lose €1 in each trial by choosing the card with the higher/lower number.”

2. You might consider noting that the correlation coefficients in Figure S3 are relatively small and depend in part on which map you are comparing it to. I bet if you included the map for the default mode, you would find a higher correlation. In any case, please state that you selected these maps to test specific a priori hypotheses, and that the correlations are relatively small. While the analysis is quantitative, the conclusions drawn from this seem qualitative – compared to other maps of interest, this yields the highest correlation coefficient, but there is no statistic reflecting whether this correlation coefficient is significantly higher than the others.

We thank the reviewer for this suggestion. Indeed, the correlation between our trustworthiness map and the default-mode network is $r = .22$, the highest correlation value among the terms we tested. Based on the reviewers’ previous comments, we selected those maps to test the associations of our trustworthiness map with specific functional roles. In particular, we wanted to show that our map was more likely associated with the social domain as opposed to the reward, congruency or risk domains. We further agree that the correlations are relatively small, but we could not test more specific functional roles such as trustworthiness or honesty (no meta-analytic maps were found for these terms) and the correlation values of our results are in line with previous research¹. We now added these considerations in the manuscript as follows (lines 679-685, page 34):

“We selected meta-analytic maps based on 12 different terms to test the specific a priori hypothesis that the trustworthiness decoding map more likely related to functional roles in the social domain as opposed to the reward, risk and congruency domains. It has to be noted that the observed correlations are relatively small but are comparable to previous research¹. Moreover, while the analysis is quantitative, the conclusions that can be drawn are descriptive in nature, as there is no inference statistics that tested whether any of the observed correlation coefficients is significantly higher than the others.”

3. A better citation for the neurosynth decoder might be Rubin et al., Plos Comp. Bio., 2017.

We thank the reviewer and added the suggested citation.

4. Your functional connectivity results agree with a recent paper that show stronger TPJ coupling with a more extensive social cognition network with increasing trust (Engelmann et al., Science Advances, 2019). It might be helpful to incorporate these supportive results in the paragraph starting on line 366.

We have now incorporated this interesting and converging evidence into the Discussion of the manuscript as follows (lines 388-393, page 22):

“Interestingly, a recent work indicates that connectivity of the left pTPJ with other social cognition regions supports behavioral trust and that an experimentally-induced disruption of trust (via aversive affect) was concomitantly followed by the suppression of pTPJ connectivity during trust decisions. These findings suggest a pivotal role of pTPJ connectivity in integration of behaviorally-relevant signal².”

5. Your analysis reported in Figure S5 and page 26 reports an interaction without a follow-up test of the interaction, so we do not know what drives the interaction. I suggest you run post-hoc tests comparing the average choice probability for honest vs dishonest advisors for cards 1-4 (averaged) and cards 6-9 (averaged). The former should yield dishonest > honest, while the latter should yield honest > dishonest to support your argument. In the absence of this test, the interaction could be driven by the difference between specific cards (e.g., 3 and 9), which would not be so interesting.

We thank the reviewer for the suggestion and added the suggested post-hoc analyses to the Methods (lines 509-517, pages 27-28):

“To test the hypothesis that this interaction effect was due to the difference in trust in the advisors and was not simply driven by differences between specific cards, we ran post-hoc t-tests and compared the average choice probability for the honest and dishonest advisors for cards 1-4 and cards 6-9. Results indicate participants were less likely to choose a card when honest advisors told them a low number was on the card (honest vs. dishonest advisors for cards 1-4: $t_{(30)} = -2.97$; $p < 0.006$) but more likely to choose a card when honest advisors told them a high number was on the card (honest vs. dishonest advisors for cards 6-9: $t_{(30)} = 2.88$; $p = 0.007$).”

Reviewer #2 (Remarks to the Author):

While the revision was responsive it also raised further questions.

We thank the reviewer for this consideration and for helpful comments that have improved the quality of the manuscript.

The advisors provided either correct or incorrect advice in every trial. This is not very realistic and could have raised doubt about the setup. Moreover, the dishonest advisor could provide more less inaccurate advice in the take advice game and be either (i) deliberately misleading or (ii) simply riskier (i.e., with regard to the decision provide similar information as the advisor but with less precision and never match the actual card). In the former case, the optimal response to the dishonest advisors probably would have been to go against the advice,

which should have led to behavior following a vertically flipped S-curve in Figure S5 rather than a less pronounced standard S-curve whereas in the latter case it may be rational to keep on relying on the dishonest advice, although this should be confirmed formally. This point should be described and clarified in the Methods (in the paragraph starting at line 475).

We are thankful for these hypotheses. The first hypothesis mentioned by the reviewer, i.e., that participants went against the adviser's advice, was indeed disconfirmed by the analyses on the card-choice probabilities (lines 496-517, pages 27-28). Had participants wanted to go against the advice, we would have expected a vertically flipped S-curve, as correctly pointed out by the reviewer. The second hypothesis mentioned by the reviewer, namely, that participants might have considered rational to keep relying on the dishonest advice, does not seem to be supported by the data either. In fact, our behavioral analyses revealed that participants were less likely to take a dishonest advice and they indeed took on average less advice from the dishonest adviser than the honest adviser (lines 115-117, page 7). Finally, the new analyses depicted in Fig. S6 (old Fig. S5) confirmed that across all possible pieces of advice in the game, participants were trusting the advice received from dishonest advisers less. These results suggest that for the same piece of advice, the likelihood someone is going to take the advice hinges on their trust in the adviser. An attempt to translate our results into a real-life scenario would be the following: You tell me to make decision X because you say this is by far the best decision I could make. Well, if I don't trust you, I'll be less convinced by your excitement and hence be less likely to make decision X. However, if someone else I trust gives me the same advice, I'll be more likely to make decision X.

Thus, we believe that our results suggest a third (and more parsimonious) hypothesis. In particular, they indicate that participants were simply *discounting* the advice of dishonest advisers. The ANOVA and new post-hoc analyses on the card-choice probabilities (Fig. S6) point to a decrease in the probability of a behavioral pattern that represented the optimal strategy in the game. This decrease in choice probability for the same piece of advice provided by dishonest advisers as opposed to honest advisers might reflect a devaluation of the advice of dishonest advisers, that is, a decrease in the recognized advice informativeness. We now describe this point and clarify it in the Methods as suggested (lines 517-523, pages 28):

“These results suggest that participants were discounting the advice of a dishonest adviser, likely because they believed it to be less informative or uninformative. In other words, this decrease in the likelihood of the use of the optimal strategy for dishonest advisers suggests a devaluation of their advice. Overall, these findings indicate that for the same piece of advice, the likelihood someone is going to take the advice hinges on their trust in the adviser or, complementary, on how much they value the adviser's advice (i.e., recognize it as informative).”

The link between the two tasks seems still somewhat overblown. Trust towards the dishonest advisers decreases already in the take advice game. It therefore seems to be more of a validation that trust towards the dishonest advisers is relatively low in the trust game (and that advice taking entails trust such that the labels of the MVPA (from the take advice game predicting subsequent trust in the trust game) might just as well have been "trust high" vs "trust low" rather than "honest" vs. "dishonest"). Accordingly, any trust-reducing behavior other than dishonesty should have yielded similar findings. These points should be acknowledged in the Discussion (e.g., in the paragraph starting at line 394).

We thank the reviewer for this suggestion. As mentioned in our previous response, the trust game was used to probe the transfer effect of the honest reputation established in the TAG on the basis of the honest advice to another context. Moreover, we never meant to argue that honesty is the only determinant of trustworthiness impressions and we apologize if we gave this impression. As requested by the reviewer, we have now emphasized these points in Methods and Discussion, in addition to already existing mentions in the Results (e.g., lines 138-140, page 8 and lines 208-210, page 13).

In the Methods (lines 531-532, pages 28):

“The TG was used to probe the transfer effect of the honest reputation established in the TAG on individual trust in a new social interaction.”

In the Discussion (lines 430-433, page 23):

“Another interesting research question for future studies relates to how other factors of trustworthiness impressions (like competence and benevolence) interact with honesty to elicit trust and/or distrust in others.”

Regarding the possibility to combine GLMs 1 and 2, I may be wrong but could there not be one GLM with one feedback regressor which is modulated by two "parametric" modulators of two levels each, one for honesty and one for gain/loss?

We are thankful for this suggestion. We agree with the reviewer that it might be possible to build one feedback regressor coding overall for feedback and modulated by two categorical parametric modulators. However, with this GLM, only one beta for honesty/dishonesty and one for gain/loss can be estimated (i.e., one for the interaction between the feedback regressor and each parametric modulator). For our multivariate analyses, two beta images were required to train the machine learning algorithms in our multivariate analyses. Thus, two separate regressors for the feedback phase had to be defined. Moreover, as pointed out in our previous response, because the same trials were used for the feedback regressors coding the advisers' honesty/dishonesty on the one hand and participants' gains/losses on the other (just differently categorized), we needed to build two separate GLMs (one for honesty/dishonesty and one for gain/loss) to be able to estimate the respective beta coefficients.

To test the robustness of our results yielded by our two separate GLMs, we ran a further control analysis using the GLM suggested by the reviewer and compared the univariate results of this GLM with the univariate results of our GLM1. As can be seen in Fig. S5., univariate results did not change using one single parametric GLM as opposed to two separate GLMs. Importantly, the two parametric modulators in the suggested GLM were orthogonalized, that is, the depicted results refer to unique variance explained by honesty after controlling for feedback-related signal. The fact that these results were virtually the same as those yielded by GLM1 strengthens our previous univariate and multivariate findings. We added this control analysis to the Results and Methods.

Results, lines 239-252, page 14:

“The definition of two separate GLMs (i.e., GLM1 and GLM2) was necessary to estimate separate beta images to train the machine learning algorithms in our previous multivariate classification and regression analyses. However, it leaves the question unanswered as to whether the observed neural signatures for honesty and dishonesty are specific to social

information processing or are confound by neural signatures of nonsocial feedback processing. We thus tested the specificity of our findings by comparing the univariate results yielded by the two separate GLMs with results of a single parametric GLM with only one feedback regressor and two categorical, parametric modulators (one coding for value information and one for the adviser's trustworthiness; see Methods). Notably, this single parametric GLM allowed us to control for spurious signal by orthogonalizing the two parametric modulators. As can be seen in Fig. S5, our results hold also with this GLM definition, suggesting that the observed neural signatures of trustworthiness are specific to social information processing."

Methods, lines 620-627, pages 31-32:

"Finally, to control that the observed neural signatures for honesty and dishonesty are specific to social information processing and are not confound by neural signatures of nonsocial feedback processing, a single parametric GLM was built with only one feedback regressor and two categorical, parametric modulators, i.e., first one coding for value information (1 = gain, -1 = loss) and then one for the adviser's trustworthiness (1 = honesty, -1 = dishonesty). The two parametric modulators were orthogonalized to be able to capture unique variance related to social and nonsocial feedback processing."

Fig. S5. Control parametric GLM analysis.

Comparison of univariate results based on GLM1 coding for honesty and dishonesty (red), and on the control parametric GLM with only one feedback regressor and a parametric modulator for honesty/dishonesty (green) orthogonalized to the parametric modulator for gain/loss. The two GLMs yielded similar results. In yellow are the overlaps depicted.

GLM, general linear model; IPL, inferior parietal lobule; DLPFC, dorsolateral prefrontal cortex; ACC, anterior cingulate cortex; VMPFC, ventromedial prefrontal cortex.

References

- 1 Gao, X. *et al.* Distinguishing neural correlates of context-dependent advantageous- and disadvantageous-inequity aversion. *PNAS* **115**, E7680-E7689, doi:10.1073/pnas.1802523115 (2018).
- 2 Engelmann, J. B., Meyer, F., Ruff, C. C. & Fehr, E. The neural circuitry of affect-induced distortions of trust. *Sci Adv* **5**, eaau3413, doi:10.1126/sciadv.aau3413 (2019).

REVIEWERS' COMMENTS:

Reviewer #1 (Remarks to the Author):

I thank the authors for addressing all my comments with such care. I have no further comments and fully support publication of this manuscript. Congratulations on this excellent contribution to social neuroscience.

Reviewer #2 (Remarks to the Author):

Thank you for a responsive revision. My previous first point (that it is not very realistic that the advisors provided either correct or incorrect advice in every trial) should be acknowledged in the manuscript together with an explanation of why the advisors should be honest or dishonest and of what the subjects were told about this.

Response to reviewers

Reviewer #1:

I thank the authors for addressing all my comments with such care. I have no further comments and fully support publication of this manuscript. Congratulations on this excellent contribution to social neuroscience.

We thank the reviewer for this appreciation of our work and for supporting its publication.

Reviewer #2:

Thank you for a responsive revision. My previous first point (that it is not very realistic that the advisors provided either correct or incorrect advice in every trial) should be acknowledged in the manuscript together with an explanation of why the advisors should be honest or dishonest and of what the subjects were told about this.

We thank the reviewer for the constructive comments. We have addressed the last points of the reviewer as follows:

That the 100% contingency in the advisers' advice-giving behavior is not very realistic (lines 527-529, page 27):

"Finally, it has to be noted that although the reputation dynamics set in motion by our design closely resemble real-life scenarios, the fact that advisers provided either correct or incorrect advice in every trial might not seem very realistic."

Why we needed advisers who were either completely honest or dishonest (lines 532-538, page 27):

"We chose a 100% contingency in the advisers' advice-giving behaviors over, e.g., a probabilistic instantiation thereof because we wanted to avoid that the multivariate algorithm for decoding neural signal related to the social character of the advisers might have picked some spurious signal evoked, for instance, by the partner's behavioral inconsistency or prediction error associated with subjective expectancies. However, future studies might want to consider introducing some variability in the behavior of pre-programmed human-like agents."

Whether subjects knew about this (lines 531-532, page 27):

"participants needed to learn trial-by-trial the behavioral conduct of each adviser, as they were not instructed about the underlying behavioral contingency"